# Amyloid-polysaccharide interfacial coacervates as therapeutic materials

Mohammad Peydayesh [1], Sabrina Kistler[2], Jiangtao Zhou [1],
Viviane Lutz-Bueno[1,3], Francesca Damiani Victorelli[1], Andréia Bagliotti Meneguin[4],
Larissa Spósito[4,5], Tais Maria Bauab[5], Marlus Chorilli[4] & Raffaele Mezzenga [1,2] ✉

Coacervation via liquid-liquid phase separation provides an excellent opportunity to address the challenges of designing nanostructured biomaterials with multiple functionalities. Protein-polysaccharide coacervates, in particular, offer an appealing strategy to target biomaterial scaffolds, but these systems suffer from the low mechanical and chemical stabilities of protein-based condensates. Here we overcome these limitations by transforming native proteins into amyloid fibrils and demonstrate that the coacervation of cationic protein amyloids and anionic linear polysaccharides results in the interfacial self-assembly of biomaterials with precise control of their structure and properties. The coacervates present a highly ordered asymmetric architecture with amyloid fibrils on one side and the polysaccharide on the other. We demonstrate the excellent performance of these coacervates for gastric ulcer protection by validating via an in vivo assay their therapeutic effect as engineered microparticles. These results point at amyloid-polysaccharides coacervates as an original and effective biomaterial for multiple uses in internal medicine.

Molecular self-assembly is a promising approach for developing advanced ordered materials by spontaneously directing individual macromolecules to assemble into a diverse set of supramolecular structures and architectures[1]. In this disorder-to-order transition driven by noncovalent interactions, many biomolecules, including protein and polysaccharide building blocks, form well-defined structures, often associated with new functionalities[2–4].

Amyloid fibrils are supramolecular protein assemblies composed of β-sheets and β-strands parallel and orthogonal to the fiber axis, respectively, resulting in highly ordered H-bonded networks[1,5]. Under certain conditions, they can be formed from animal-based (e.g., β-lactoglobulin[6], lysozyme[7], and bovine serum albumin[8]) and plant-based (e.g., soy[9], pea[10], and oat[11]) protein monomers through unfolding, hydrolysis, and aggregation steps. Owing to their unique surface functionality and a high surface-to-volume ratio[12], amyloid fibrils are an

ideal candidate to develop advanced green materials for a wide range of applications[13–17]. Furthermore, they can be obtained sustainably and cost-effectively by valorizing agri-food industry waste[18], such as whey[19], proteins from oilseed meals[20] and marine fisheries[21].

Recently, complex coacervation has gained traction in the supramolecular self-assembly of biomacromolecules[22]. Coacervation is an associative liquid-liquid phase separation (LLPS) phenomenon that occurs through electrostatic interaction between oppositely charged macromolecular building blocks, such as proteins, polysaccharides and polymers in solution, resulting in a colloidal-rich phase, the coacervate, and a remaining biomacromolecule-depleted phase in equilibrium with the denser coacervate[22,23]. The reduction in free electrostatic energy of the system caused by the interaction of oppositely charged polyions and the release of counterions is the primary driving force for coacervation[24], despite the mixing entropy effects,

[1]ETH Zurich, Department of Health Sciences and Technology, 8092 Zurich, Switzerland. [2]ETH Zurich, Department of Materials, 8093 Zurich, Switzerland. [3]Paul Scherrer Institute PSI, 5232 Villigen, Switzerland. [4]Department of Drugs and Medicines, School of Pharmaceutical Sciences, São Paulo State University, 14800-903 Araraquara, Sao Paulo, Brazil. [5]Department of Biological Sciences, School of Pharmaceutical Sciences, São Paulo State University, 14800-903 Araraquara, Sao Paulo, Brazil. ✉e-mail: raffaele.mezzenga@hest.ethz.ch

which tend to disperse them[25]. LLPS and coacervates can eventually undergo aging and lead to a liquid-to-solid transition (LST) and precipitation[26]. Although coacervation can take place in the absence of precipitation, precipitation induced by a pH change is always preceded by coacervation[27]. To date, the application of coacervates has been considered in food[23,28], personal care products[29,30], cellular transfection[31] and interaction motifs in biology[22,32]. More specifically, protein-polysaccharide complexes have been used in a wide range of biomedical applications to provide specific biological functions, including nutraceutical and drug delivery platforms[33], biomimetic adhesives[34], sensors[35], and artificial tissues or implants for tissue regeneration[36]. However, the application of protein-polysaccharide coacervates is restricted due to the low mechanical strength and chemical stability of native proteins, particularly when high mechanical stresses are involved. To improve coacervates' stability and retain their desired structures, crosslinking with aldehydes, such as glutaraldehyde, has been proposed as a possible improvement[37]. However, the biomedical use of aldehydes crosslinkers has been linked to a number of negative health effects in humans[38], so that its use in vivo is challenging, particularly via oral administration. We hypothesize, therefore, that a possible way forward in this field may be offered by transforming physically rather than chemically the proteins, such as converting them into robust artificial amyloid fibrils, which are known for their excellent biocompatibility and activities for biomedical applications[39].

Herein, we show how the coacervation of protein amyloid fibrils and polysaccharides results in high-order molecular self-assemblies for designing advanced materials, including films, fibers, and capsules, with properties by far higher than those of the homologue coacervates obtained by the same constituents but with the protein in its native form. To do so, semiflexible amyloid fibrils (AF) from β-lactoglobulin, the major protein in whey (a by-product of the dairy industry), are used as polycation to coacervate with hyaluronic acid (HA), a long-chain linear natural polysaccharide, as polyanion. We develop the coacervation phase diagrams in different conditions and systematically characterize the hierarchal structures of the ensued materials. We finally demonstrate their application for gastric ulcer protection via an in vivo study to illuminate the potential of these coacervates in therapeutics and medicine. In addition, to show the generality of the approach, we present in the SI one further application example of coacervates beyond therapeutics, specifically designed for water purification applications.

## Results and discussion
### AF-HA coacervates
Figure 1 shows a schematic of different strategies for producing versatile materials upon the AF-HA coacervation. When the AF solution was uniformly layered on top of the HA solution (in an equal volume ratio), coacervation occurred, and a membrane film formed immediately at the interface between the two solutions (Fig. 1a). The order in which the two solutions were combined (HA over the AF) could be changed; however, due to the higher viscosity of the HA, uniformly spreading it on top of the AF solution to achieve a homogeneous membrane film was more difficult. The effect of coacervation time on the thickness of the resulting films is shown in Supplementary Fig. 1. The film thickness exceeds $20\,\mu m$ within the first 2 h of coacervation; after that, no significant difference in film thickness was observed. Alternatively, a continuous fiber can be formed by pulling the interface formed by extruding AF inside HA. The interface string is continuously rebuilt by diffusion and assembly of both components during this wet spinning process, resulting in fibers of virtually infinite length (Fig. 1b). Eventually, as observed in Fig. 1c, closed capsules can also be produced instantly by injecting one solution directly into the bulk of the other, resulting in either AF-filled or HA-filled capsules. This can be an asset in the encapsulation

application, where the targeted compounds can be incorporated with a more compatible inner phase.

Supplementary Fig. 2 exhibits the bulk coacervation of AF and HA, where the coacervate and dilute phase (equilibrium solution) are clearly distinguishable. The equilibrium solution was transparent and had a very low viscosity, different from the two precursors. Figure 2a–c visually show the developed coacervate materials, including films, fibers and capsules, respectively. As shown in Fig. 2a, the films are optically transparent and tailorable in different sizes and shapes. The mechanical stability of the films was evaluated by obtaining the stress-strain curve (Supplementary Fig. 3). The films exhibit good mechanical stability and low stretchability, with a maximum stress of 21.9 MPa, elongation of 4.3%, and Young's modulus of 6.1 MPa. Moreover, as observed in Fig. 2b and Supplementary Movie 1, by extruding one droplet of AF solution in the bulk of HA and then picking up and withdrawing the formed interface, a uniform fiber with a length of 1.2 m could be spun and dried. The thickness of the resulting dry fibers was $20 \pm 2\,\mu m$, as shown in the optical microscopy, atomic force microscopy (AFM), and scanning electron microscopy (SEM) images in Supplementary Fig. 4.

The obtained capsules were homogenous, robust and round in shape and could be transferred to other targeted media (Fig. 2c). The capsules can be formed in versatile sizes by adjusting the volume of the injected inner solution. The colorful image in Fig. 2c, obtained with the LC (liquid crystal)-PolScope device, depicts the direction of the fibrils in the plane; see the colormaps in the top-right corner. The structure, texture, and stability of coacervates and the entrapped material are governed by the strength of intermolecular interactions between AF-HA[24,40]. In sharp contrast with the good physical and mechanical properties featured by AF-based coacervates, the weak coacervation between native β-lactoglobulin proteins and HA (Supplementary Fig. 5) yielded only poor coacervates: the formation of uniform and resilient films was not possible, fibers could not be spun beyond 1 cm, and the capsules had only irregular and loose shapes, highlighting the central role of amyloid fibrils in controlling the final properties and structure of coacervates.

To shed light on the interfacial AF-HA self-assembly, we investigated the effect of biopolymers' mixing ratio and pH on the coacervation. Figures 2d and 2e depict the concentration phase diagram of coacervation for producing AF-filled or HA-filled capsules, respectively (the related photographs are presented in Supplementary Figs. 6 and 7). The phase diagrams are divided into four regions: No capsules, unstable capsules, stable capsules, and gelation of one of the precursors. No visible coacervates were observed in low concentrations and AF/HA ratios, and only intrapolymeric complexes were formed[41]. By increasing the concentration of AF and HA, the phase diagrams enter the unstable and later stable coacervates region. Since at each pH, there are specific concentration ratios for which complex electroneutrality is achieved[41], the amount of AF available per HA chain is of great importance. The biopolymers ratio affects the amount of complex formation by influencing the charge balance between proteins and polysaccharides[23]. It is evident from the concentration phase diagrams that at the studied pH of 3, a minimum concentration of around 1 wt.% for AF and 0.5 wt.% for HA is required for successful coacervation. The same coacervation behavior was observed when developing fibers and films, as shown in Supplementary Figs. 8–11. However, in all cases, at HA concentrations above 2 wt.% HA gelation occurred, impeding the coacervation process. The molecular weights of the polysaccharides involved in protein coacervation have also been reported to play a role. High molecular weight polysaccharides, such as HA, are desired for coacervation because low molecular weight polysaccharides interact by ion pairing, inhibiting the coacervation phenomenon[24].

Another critical factor in the coacervation of AF and HA is pH, which directly determines the charge density of polyions and, thus, the

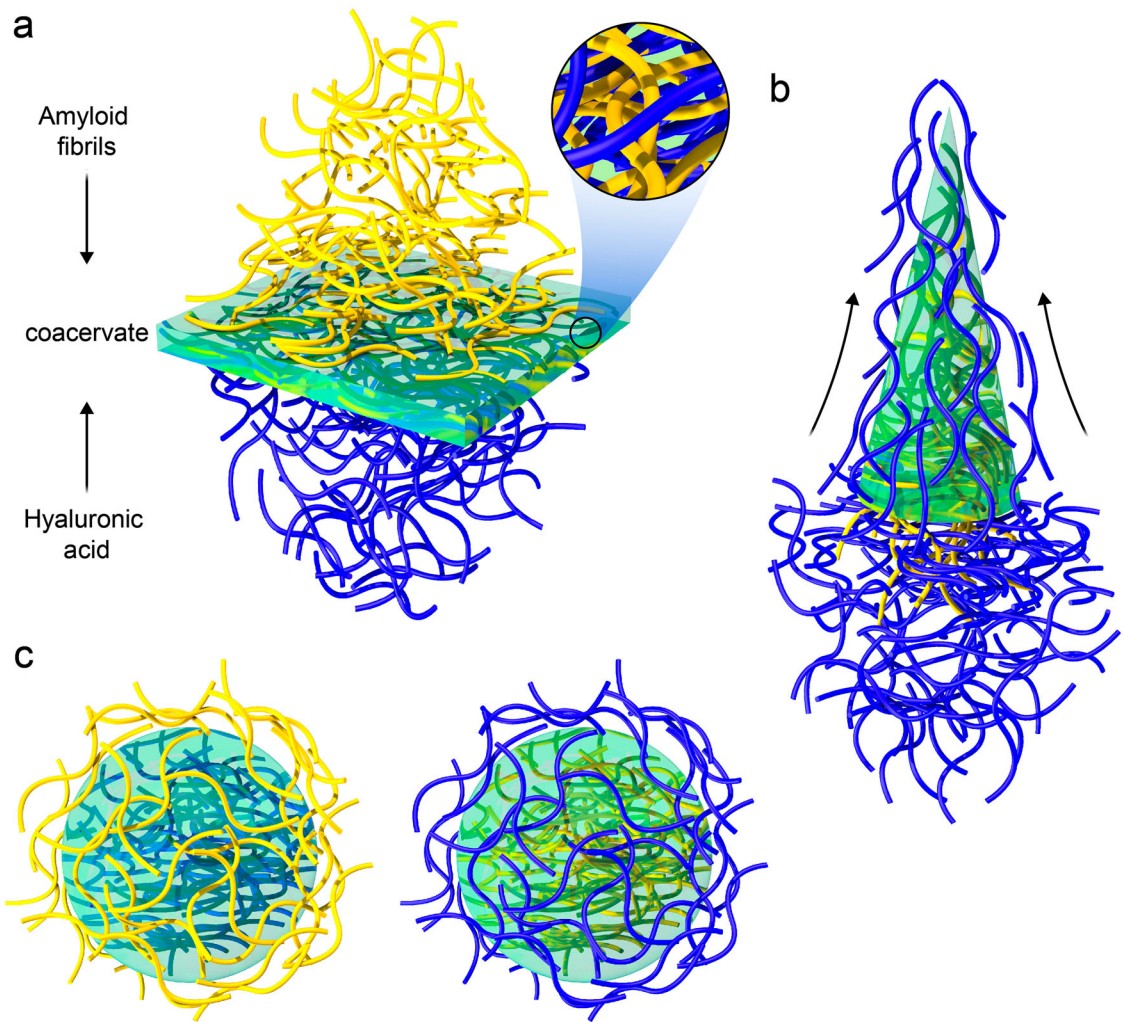

**Fig. 1 | Schematic of AF and HA interfacial coacervation for developing advanced materials. a** films, **b** fibers and **c** capsules.

electrostatic forces[25]. Supplementary Fig. 12 depicts the electrophoretic mobility of HA and AF over a wide pH range. Although HA has a net negative charge throughout the pH range, AF have an isoelectric point (IEP) of 5.2 and a positive charge at pHs below this value.

As shown in the pH phase diagrams, well-formed stable capsules were achieved when producing AF-filled (Fig. 2f) or HA-filled (Fig. 2g) coacervates (the related photographs are presented in Supplementary Figs. 13 and 14), at acidic pHs below the IEP of AF. In acidic pH, positively charged AF electrostatically interacts with negatively charged HA, resulting in net neutrality and efficient coacervation[42]. However, at pHs higher than the IEPs AF, the net charge of the fibrils shifts to the negative, resulting in the formation of weak and unstable coacervates. Capsule formation was eventually impossible at pH around the IEP due to AF gelation. To clearly reveal that the LLPS occurred due to electrostatic interactions (complex coacervation), we performed the bulk coacervation of AF and HA at pHs 9 (both AF and HA are negatively charged) and 3 (AF has a positive charge and HA has a negative charge). As observed in Supplementary Fig. 15 at pH 9 the AF-HA mixture is miscible, translucent and no gel was formed, but by decreasing the pH to 3 the coacervation between oppositely charged precursors occurred and resulted in robust, turbid gel, highlighting the predominate role of electrostatic interaction in this phenomenon.

Another proof of complex coacervation (electrostatic interaction) is the sensitivity to the solution ionic strength. To evaluate the ionic strength effect, the bulk coacervation of AF and HA was done in the presence of salt (NaCl) at different concentrations, i.e., 0, 10, 100, and 1000 mM. Supplementary Fig. 16 shows the visual appearance of coacervation samples as well as their turbidity as a function of salt concentration. As observed, strong coacervation occurred in the absence of salt, resulting in a coacervate phase and a transparent equilibrium solution. The turbidity of the coacervate phase was around 2000 a.u. By increasing the salt concentration, charge screening of precursors takes place, and coacervation becomes weaker. At the salt concentration of 1000 mM, the sample gel properties were lost, and homogenous and cloudy solutions with turbidity around 500 a.u were obtained.

It is possible, of course, to initiate the coacervation with anionic AF (at pHs above 5.2) and cationic polysaccharides such as chitosan (CS). To show this possibility, we ran the coacervation with these two precursors at pH 8. As observed in Supplementary Fig. 17, it is possible to produce AF sacs in the CS phase or CS sacs in the AF phase, as well as an interfacial film after layering them on top of each other. Figure 2h–k show confocal microscopy images of the capsule's wall, including rhodamine-labeled AF, the fluorescein-labeled HA, merged (corresponding to the combined signals from labeled AF and HA), and transmission images, respectively. These images clearly reveal the incorporation and colocalization of both AF and HA within the capsule's wall upon the coacervation.

We further elucidate the structure and composition of coacervates by characterizing the films via various methods. As explained earlier (Fig. 1), AF was placed on the surface of HA to produce films, so that the top surface of the resulting coacervate film

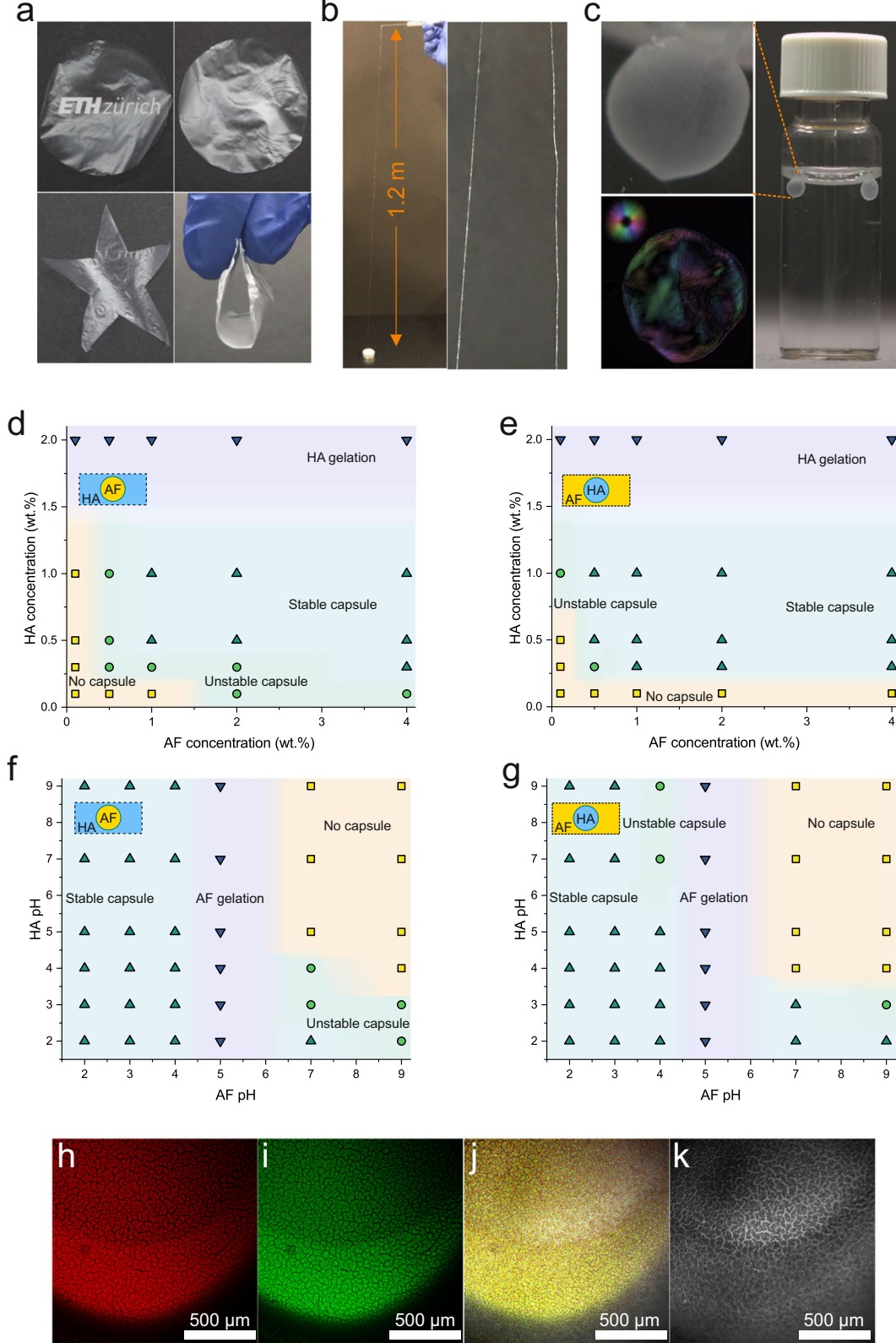

**Fig. 2 | Various amyloid fibrils (AF) - hyaluronic acid (HA) coacervate products and the effect of major parameters on coacervate capsules.** Photographs of **a** films, **b** fibers and **c** capsules formed by coacervation of AF and HA (The colorful image in panel c shows the direction of the fibrils in the plane, see the colormaps in the top-right corner). **d** Concentration phase diagram for producing capsules (pH 3). AF as the inner phase and HA as the outer phase and **e** HA as the inner phase and AF as the outer phase. **f** pH phase diagram for producing capsules (AF 2wt.% and HA 1wt.%). AF as the inner phase and HA as the outer phase and **g** HA as the inner phase and AF as the outer phase. Yellow square and shading refer to no sac, green circle and shading refer to unstable sac, blue up-triangle and shading refer to stable sac and purple down-triangle and shading refer to AF/HA gelation. Confocal microscopy of **h** rhodamine-labeled AF, **i** the fluorescein-labeled HA within the capsule wall, **j** merged image (confirming colocalization of the AF and HA within the capsule's wall) and **k)** transmission image. The confocal microscopy experiment was repeated at least three times and representative images were shown.

was in contact with AF, and the bottom side was in contact with HA. To characterize our film's morphological, chemical and biophysical properties at the nanoscale, we applied AFM infrared nanospectroscopy (AFM-IR)[43], as schematically shown in Fig. 3a. We obtained the AFM morphological images, the map of the infrared absorption at 1654 cm⁻¹, and the stiffness map on both sides of the film (Fig. 3b). Morphological images show the AF network on the top and a smoother but porous surface on the bottom of the film (more details can be seen in conventional AFM in Supplementary Fig. 18). Specifically, IR absorption and stiffness maps proved that both sides of the film have a uniform chemical and compositional distribution, as well as a homogenous biophysical property. Furthermore, infrared spectra were collected at the indicated location on the AFM images (Fig. 3c). On the top side of the film, we observed three main absorption peaks, at 1654, 1552, and 1400 cm⁻¹, precisely located in distinct regions of amide I, II, and III, respectively[43,44]. In particular, the absorption peak in the region of amide I (1700–1600 cm⁻¹) is attributed to the C=O stretching vibration associated with protein secondary and quaternary structure[43,45]. In this case, the main peak and shoulders in the amide I indicate the rich composition of β-sheet and random coil on the top surface of the film. In comparison, although the IR absorption spectrum of the bottom side of the film shared similarities with the top side, significant differences can be found. First, we observed the absorption peaks of amide I, II, and III, close to the spectrum of the top side (The wavenumber of the amide I peak remains identical to the top side), confirming the existence of AF also on the bottom side of the film. Yet, we also noticed the fingerprints of HA characteristic bands: the major difference located at 1037 cm⁻¹, associated with the C-O-C hemiacetalic system saccharide units[46,47]. The shoulder peaked at 1617 cm⁻¹, and the rise of the peak at 1401 cm⁻¹ are correlated with symmetric and asymmetric vibration of carboxyl (−COOH) groups, while the prominent peak at 1728 cm⁻¹ corresponds to the stretching −COOH groups of HA structure[47]. This nanoscale evidence from localized AFM-IR results indicates that while AF dominates the top side of the film, they are also capable of diffusing to the bottom side, which includes both AF and HA.

A representative SEM image of the top surface of coacervate films indicates the complete coverage of the surface by AF (Fig. 3d), while Fig. 3e depicts a cross-section image of a vertically aligned coacervate film (top layer on the left, bottom layer on the right), with the AF-HA coacervate sharp interface clearly visible. This self-assembled interface zone (perpendicular orientation compared to the top and bottom sides of films) is formed by coacervation caused by rapid electrostatic interaction between AF and HA. As observed in Fig. 3f, the bottom surface of the coacervate film has a more random spongy morphology (similar to other reports on HA-based coacervate films[48]), yet some AF can be seen on this side (see inset).

The surface chemistry of coacervate film was further characterized by X-ray photoelectron spectroscopy (XPS), by comparing pure AF and HA, as well as the top and bottom of coacervate films (Fig. 3g). The carbon (C1s) and Oxygen (O1s) signals were clearly detected. As observed, AF spectra indicate the presence of functional groups such as C-C, C-H, C=C, O-C=O, and N-C=O, while HA spectra show the C-O-R group[49]. By comparing the XPS spectra of two sides of the film, it becomes clear that the two sides of the coacervate film do not exhibit an identical pattern. The spectra of the top layer of the film and the reference AF are very similar, indicating the surface of this side consists of essentially amyloid, with little/no HA (at least within the 10 nm depth probed by XPS). On the other hand, the spectra of the bottom layer can be reasonably interpreted as a combination of AF and HA references, i.e., this side contains a contribution from both materials.

The film cross-section is measured by in-plane wide-angle X-ray scattering (WAXS) of samples containing pure AF, and the AF+HA coacervates. The fingerprint of AFs is their cross-β structure, in which

protein β-sheets lay parallel to the fibril axis, while the β-strands within these sheets organize themselves perpendicular to the fibril axis (Fig. 3h). The intensity of 2D scattering patterns ($I$) was integrated as a function of the azimuthal angle ($\chi$), which provides information about the structural organization and anisotropy of the film's components (Fig. 3j). The results confirm that the typical overall alignment, and cross-β scattering pattern[50,51], of the pure AF films is absent when coacervates are formed with HA, in AF+HA (Fig. 3i), indicating that the coacervation is associated with an isotropic angular distribution of the amyloids, as one would expect in a molecular coacervate. The intensity of the rings in the 2D scattering pattern is integrated as a function of the scattering vector (**q**). The peak $p_1$ is related to the distance between β-sheets ($p_1$), which varies depending on the side-groups of the proteins and their packing into AFs and is observed for both films AF and AF+HA. Peak $p_2$ arises from hydrogen bonds between two consecutive peptide backbones of the β-strands in AF and is observed for both films AF and AF+HA, as well as for the HA powder (Fig. 3k). The dimensions ($d$) related to these WAXS reflection peaks are on average about $d_1 = 9.7$ Å and $d_2 = 4.4$ Å for the AF+HA, which agree well with the average distances for pure AF films of $d_1 = 10.11$ Å and $d_2 = 4.45$ Å (Supplementary Tab. 1).

## Therapeutic application of AF-HA coacervates

We finally explore the use of obtained coacervates as a biomaterial for therapeutic applications, more specifically, gastric ulcer protection. Ulcers are caused by a gastrointestinal disorder that occurs when there is an imbalance between aggressive factors (secretion of HCl and pepsin) and factors that defend the gastric mucosa, such as secretion of bicarbonate and mucus, and production of prostaglandins, causing superficial damage that is generated by an inflammatory process that promotes tissue loss on the mucosal surface. HCl and pepsin are the main aggressors of the gastric mucosa, and their damage can be minimized or avoided by the protective factors of the gastric mucosa, such as the presence of an intact layer of gastric epithelial cells, a protective layer of mucin, secretion of bicarbonate that neutralizes the acidity present in the gastric mucosa, presence of growth factors, angiogenic factors and the presence of prostaglandins[52]. Prostaglandins decrease gastric secretion, increase mucosal blood flow and regulate gastric mucin synthesis, stimulate mucus and bicarbonate secretion, induction of angiogenesis, and various other functions to restore mucosal integrity, playing a pivotal role in ulcer healing process[53]. Studies on this disease are relevant since it is considered the most common chronic disease of the upper digestive tract in adults, affecting about 4 million people per year worldwide[54]. There are several etiological factors linked to the cause of these diseases, such as smoking, stress, nutritional deficiencies, abusive and prolonged use of drugs such as non-steroidal anti-inflammatory drugs (NSAIDs), alcohol, and H. pylori infection[55]. However, the most common factor is associated with the intake of NSAIDs, which are among the most-used drugs in the world[56,57]. In this sense, this protocol using indomethacin, an NSAID to induce gastric ulcers, is important in the study of this disease. We performed in vivo assay to verify the gastric ulcer protective properties of the coacervate microparticles using Indomethacin-induced gastric ulcer protocol (Fig. 4a). Indomethacin-related gastrointestinal toxicity occurs by inhibiting the pathway of cyclooxygenase 1 and 2 (COX-1 and COX-2) enzymes, which are responsible for the synthesis of prostaglandins. In addition, after the formation of the lesion in the mucosa, NSAIDs inhibit the first restitution events necessary to repair the initial superficial damage, as well as subsequent events, acting in the reduction of epithelial cell proliferation and in the reduction of angiogenesis, promoting a delay in the ulcer healing[58]. The microparticles were obtained by spray drying the AF-HA coacervates[22,59]. The field emission gun-SEM (FEG-SEM) images (Fig. 4b) confirmed the development of microparticles close to the spherical shape commonly formed by the spray dryer process.

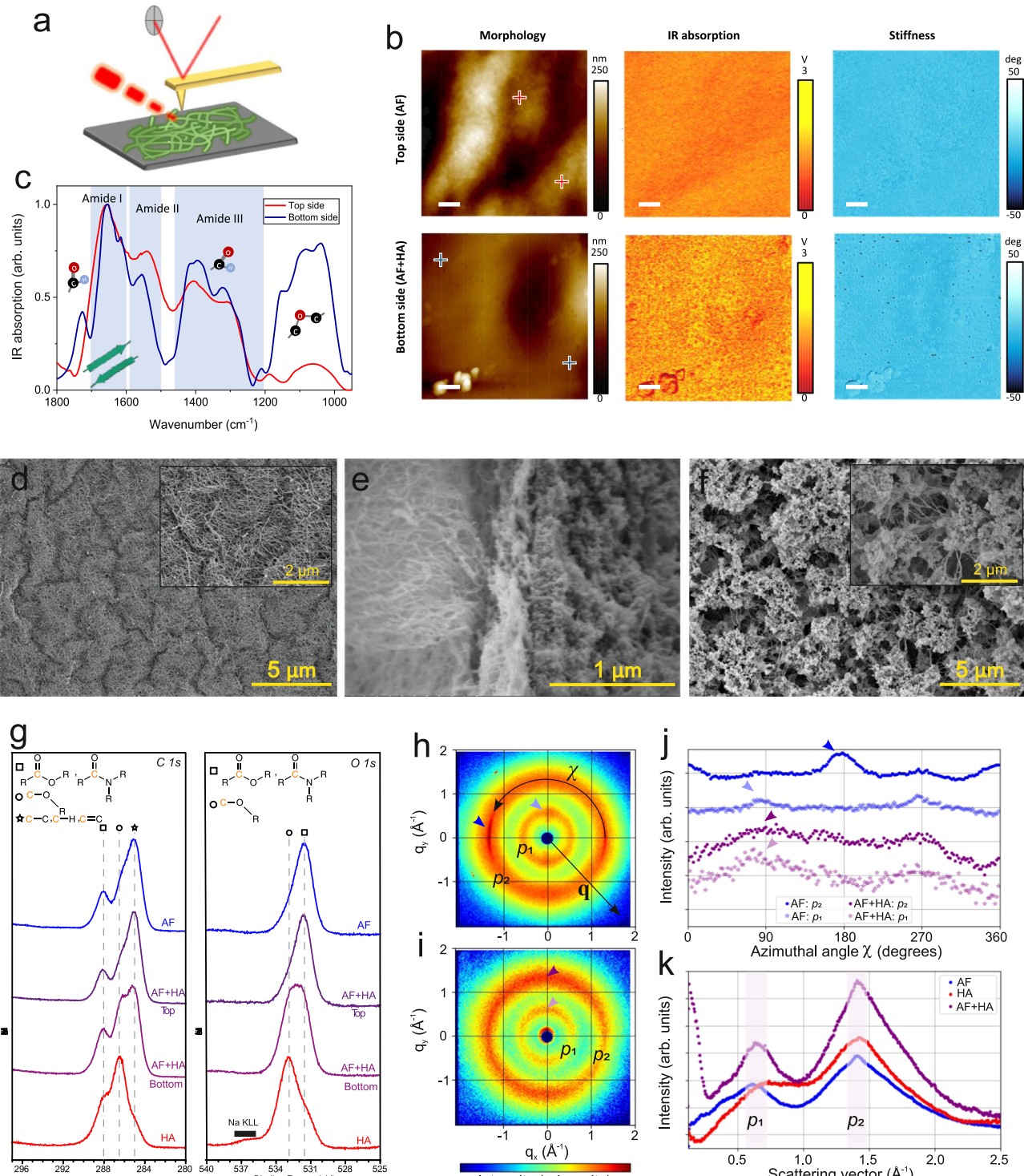

**Fig. 3 | Characterization of amyloid fibrils (AF) - hyaluronic acid (HA) coacervate films at different sides and interfaces. a** The scheme of AFM-IR experimental setup while scanning on the film. **b** Morphological images (left), maps of infrared (IR) absorption in the Amide I at 1654 cm⁻¹ (middle) and stiffness that indicates the nanomechanical properties of the top (upper) and the bottom (lower) side of the film. The crosses refer to the locations for performing IR spectroscopy measurements. The scale bar is 500 nm. The AFM-IR experiment was repeated at least three times and representative images were shown. **c** Nanoscale localized infrared spectrum of the top and bottom side of the film collected at the crossed location in panel **b**. SEM images (**d**) surface of AF side, (**e**) AF-HA interface and **f.** surface of HA side. The SEM experiment was repeated at least three times and representative images were shown. **g** XPS *C1s* and *O1s* chemical state spectra of pure AF and HA, as well as top and bottom surfaces of coacervate films. 2D WAXS scattering patterns of (**h**) pure AF and (**i**) AF + HA films. **j** Intensity *I* as a function of the azimuthal angle (χ) of AF and AF + HA films. **k** Intensity *I* as a function of scattering vector **q** of AF and AF + HA films, as well as HA powder. $p_1$ and $p_2$ are the two identified peaks.

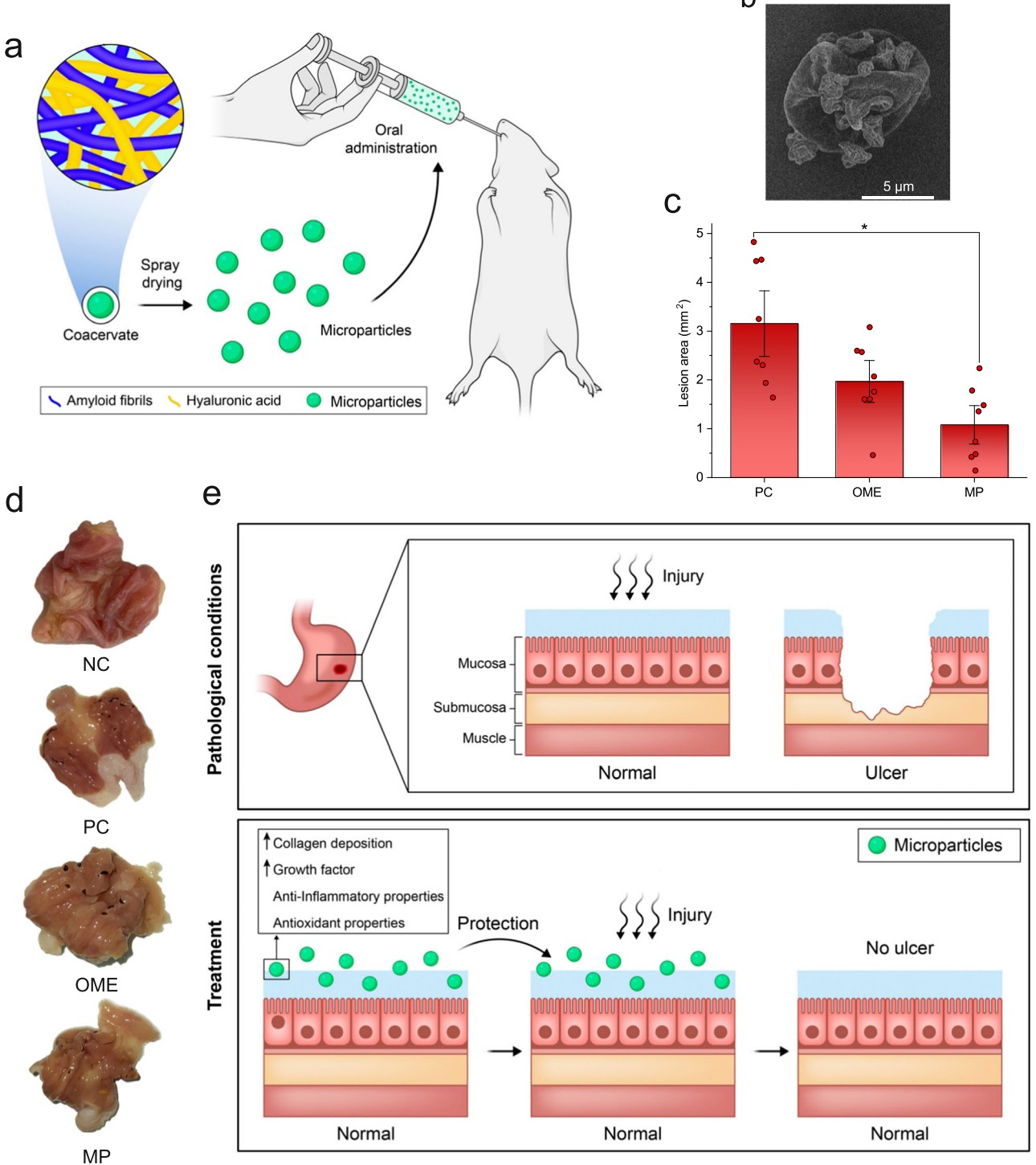

**Fig. 4 | AF-HA coacervate therapeutic application. a** Schematic representation of the development of microparticles and their application in the in vivo assay. **b** FEG-SEM images of spray-dried AF-HA microparticles. The FEG-SEM experiment was repeated at least three times and a representative image was shown. **c** lesion area of the gastric ulcer in rats and (**d**) macroscopic photo images of representative stomach tissue with or without gastric ulcer. Data are expressed as the mean±SD ($n = 8$); a one-way analysis of variance (ANOVA), followed by Fisher's LSD test, $p = 0.0003$, compared with the positive control group. NC negative control; PC positive control; OME omeprazole; MP microparticles. **e** Schematic of mechanisms for gastric ulcer protection by AF-HA microparticles.

The shrinking appearance is due to the contraction of the microparticles caused by fast drying during the spray drying process[60]. As observed in Fig. 4c, in the in vivo assay, the group treated with AF-HA coacervate microparticles showed the development of lesions with a significantly smaller area compared to the positive control ($p = 0.0003$), and even better than in the treatment with omeprazole, a conventional gastric ulcer protective drug, very effective in the treatment of stomach lesions, but also known for severe adverse effects (the photographs of representative stomach tissue with or without gastric ulcer are shown in Fig. 4d). The main mechanisms for gastric ulcer protection by AF-HA microparticles are schematically depicted in Fig. 4e. The main protective effect is attributed to the presence of both

HA and AF. HA presents antioxidant properties and is involved in several physiological processes, including wound repair and regeneration, morphogenesis, and matrix organization[61]. The antioxidant property of HA is related to its hydroxyl functional groups that can absorb reactive oxygen species and avoid gastric mucosal damage. Additionally, HA can promote cell proliferation, increase growth factor, and stimulate fibroblasts to produce collagen fibre contributing to wound healing[62,63]. Moreover, β-Lactoglobulin possesses anti-inflammatory and peripheral antinociceptive activities[64]. It is a rich source of sulfur amino acids such as cysteine that can stimulate the synthesis of glutathione, one of the compounds responsible for the anti-inflammatory and antioxidant activity in various tissues and organs[65]. To show that the β-Lactoglobulin AF are digestible and similar anti-inflammatory effects can be expected from them, we performed the in vitro digestion of AF-HA capsules containing AF as an inner phase. The digestion was done at 37 °C for 2 h using fresh simulated gastric fluid (SGF) containing pepsin (as a control, the same experiment was done with Milli-Q water). As observed in the control sample in Supplementary Fig. 19, after 2 h incubation at 37 °C, the long AF were detected in the water by AFM, confirming the presence and release of AF from the coacervate capsule to the bulk of water. However, for the sample treated with SGF, the AFM image did not show any evidence of the existence of AF but short fragments, indicating the complete digestion of AF. As HA has adhesive properties[66], the hypothesis that we suggest for the protective effect of microparticles is that they can adhere to the mucosa, and then HA and AF may synergistically contribute to the ulcer gastric protective properties of the coacervate microparticles, highlighting them as systems with efficient biological activity for gastric ulcer protection and possible therapeutic use.

In summary, we demonstrated how the hierarchical multiscale self-assembly via coacervation of protein amyloid fibrils and polysaccharides results in easily tunable functional capsules, films and fibers with properties far higher than those obtained using native monomeric proteins in the coacervates. The study and understanding of the various factors controlling the electrostatic interaction between the amyloid and polysaccharide building blocks, provides valuable insight into the mechanisms that govern the coacervation and determine the final structure of the electrostatic complexes. The final materials feature higher performance in gastric ulcer protection thanks to the anti-inflammatory and antioxidant activity of both HA and AF, although we believe that these materials may have a scope well beyond the therapeutics, expanding to scaffolds for tissue engineering, biomaterial templates and membranes for molecular separation.

## Methods

### Materials
β-lactoglobulin was purified from whey protein isolate (Fonterra, New Zealand) and used to prepare AF. HA sodium salt from *Streptococcus equi*, Curcumin from *Curcuma longa*, Rhodamine B, Fluorescein, Acid Fuchsin, Pepsin from porcine gastric mucosa (P6887), Calcium chloride dihydrate, Sodium chloride, and gold (III) chloride trihydrate (≥99.9%) were purchased from Sigma Aldrich.

### AF preparations
β-lactoglobulin was isolated from whey by precipitation of α-lactalbumin at pH 3.6 and 50 °C followed by a dialysis step against Milli-Q water (membranes with 6–8 kDa) for removing salts. The obtained purified β-lactoglobulin was freeze-dried and kept at 4 °C[67]. Then, AF were formed by fibrillation of purified β-lactoglobulin (2–4 wt.%) in an aqueous medium at 90 °C and a pH of 2. The conversion rate from monomer to AF for β-lactoglobulin is around 80%[68]. The presence of β-lactoglobulin AF was confirmed by birefringence using cross-polarized light and AFM (Supplementary Fig. 20). They

have diameters and lengths ranging from 2 to 8 nm and 5–10 μm, respectively.

### AF-HA coacervation
For performing the coacervation, various AF solutions (0.1–4 wt.% and pH of 2–9) were treated with different HA solutions (0.1-2 wt.% and pH of 2–9). AF and HA at concentrations above 4 and 2 wt.%, respectively, become very viscous, hampering their efficient use in the coacervation process. The capsules were obtained by extruding a droplet of one phase into the bulk of another phase in various volumes. The films were formed by smearing 1 ml of AF phase on top of 1 ml of HA phase. To prepare the fibers, 10 μL of AF solution was first extruded into the bulk of HA, and then the interface was pulled vertically by the top of the syringe needle.

### Characterization
The PolScope image was captured using a microscope (Zeiss Axio Imager M1m) equipped with LC (liquid crystal)-PolScope and 10× (Plan Neofluar) objective. An upright microscope (Zeiss AxioImager.Z2) with a ×10 objective was used to collect the polarizing optical microscopy image of the fiber. The fluorescence images were captured using a confocal microscope (Leica TCS SP8) with a 10× objective (0.75NA HC PLAN APO CS2). The surfaces and cross-sections of coacervate capsules and fibers were analyzed using SEM (Hitachi Su5000). For SEM, the structure of capsules was fixed by a solution containing 2% glutaraldehyde, 3% sucrose, and phosphate buffer at pH 7.4. Then the samples were dehydrated by solvent exchange against increasing concentrations of ethanol. Finally, the ethanol was removed by critical point drying to preserve the internal structure of the coacervate layer. FEG-SEM images of coacervate microparticles were obtained using the JEOL JSM-7000 F microscope operating at 2 kV. The electrophoretic mobility of AF and HA was measured using a Malvern Nano-Zetasizer. The mechanical stability of coacervate films was characterized by analyzing stress-strain curves obtained using a Zwick Z010 (Zwick-Roell) equipped with a 10 N load cell. The AFM-IR measurements were performed on a nanoIR2 device (Anasys Inc., Bruker), equipped with a tunable infrared quantum cascade laser (MIRcat-2400). The imaging and analysis were conducted in an enclosed environmental chamber at a low humidity condition (RH < 3%) and temperature constant of 23 °C. The morphological, IR absorption and stiffness maps were collected in the tapping mode by using a silicon gold-coated probe (PR-EX-TnIR-A) with a spring constant of 1–7 N/m and resonant frequency of 75 ± 15 kHz. A scanning rate of 0.2–0.8 Hz was applied to acquire the image size of 4 μm by 4 μm with a resolution of 512 × 512 pixels. The spectra were collected by locating the probe apex at the desired location on the AFM image of the top of the film. The spectra were recorded in the range from 900 cm$^{-1}$ to 2000 cm$^{-1}$, with a sampling resolution of 2 cm$^{-1}$. The spectra were analyzed with the built-in software (Analysis studio, Anasys, Bruker) and OriginPro. Conventional AFM measurements were performed by a Bruker multimode 8 scanning probe microscope (Bruker, USA) with an acoustic hood to minimize vibrational noise. The AF-HA fiber and the film were immobilized on the mica surface. The imaging was operated in soft tapping mode under ambient conditions, using a commercial triangular silicon nitride cantilever (Bruker, USA) at a vibration frequency of 70 kHz and a spring constant of 0.4 N/m, and a relatively soft tip-sample interaction was applied during the scanning. The surface chemical characterization of the pure samples and coacervate films was acquired by XPS using a Quantum 2000 X-ray photoelectron spectrometer (Physical Electronics, Minnesota, United States), equipped with an Al Kα monochromatic source (1486.6 eV), a hemispherical capacitor electron-energy analyzer, and a 16–channel plate detector. The samples were prepared by mechanically embedding them into thin indium foil, and all the spectra were obtained with an emission angle of 45°, in

fixed analyzer transmission mode, using a nominal X-ray beam-spot size of 150 μm and at a pressure lower than $4 \times 10^{-7}$ Pa.

Solutions of AF (2 wt.%) and AF (2 wt.%) + (HA 1 wt.%) were dried into thin films. A thin cross-section was placed on a tape with a thickness parallel to the X-ray beam; thus, the scattering intensities (I) were averaged over the whole film thickness. WAXS experiments were performed using a Rigaku MicroMax-002+ micro-focused beam (4-kW, 45-kV, 0.88-mA). The copper K-α energy with $\lambda\_e = 1.54$ was collimated by three pinholes onto a beam size of 700 μm. A single wire was vertically placed in front of the beam, and the scattered intensity was detected by a Fuji Film BAS-MS 2025 imaging plate. From the calibrated and normalized scattered intensities, we calculated the radial and azimuthal integrations. The scattering intensity in the range of scattering vector $\mathbf{q} = 0.1–2.5$ Å$^{-1}$ was radially integrated as a function of the azimuthal angle (χ), and provides information about the alignment of AF and HA in the film thickness. For the scattering curve of intensity as a function of $\mathbf{q}$, the two peaks, identified by $p_1$ and $p_2$, are a contribution from all components forming the film, and provide insights about the structural organization. Their spacings are defined as $d = 2\pi/q$, and provide the main distances of the samples.

### in vitro digestion of AF-HA coacervate

The in vitro digestion of AF-HA coacervate was applied to simulate the physiological gastric digestion process by following the INFOGEST standard protocol[69]. Briefly, fresh SGF containing the pepsin from porcine gastric mucosa (P6887) with the enzyme activity of 500 U/mL was prepared and equilibrated to 37 °C before the digestion test. One AF-HA coacervate capsule was digested in 3 mL of SGF solution, followed by the addition of CaCl$_2$ solution (0.3 M) into the digestion solution reaching a final concentration of 0.15 mM prior to digestion. The digestion was carried out in a water bath (VWR 462-0493) at 37 °C under shaking conditions (60 rpm) for 120 min, and AFM imaging was performed immediately after digestion. The control experiment was performed using Milli-Q water instead of an SGF solution.

### Indomethacin-induced gastric ulceration in vivo assay

For gastric ulcer protection assessment, we developed the coacervate microparticles using 2 wt.% of AF pH 2 and 1 wt.% of HA pH 4. The AF solution was added to the HA solution with a volume ratio of 1:0.25. The mixture was agitated with a magnetic stirring for 30 min. Then, we broke the formed bulk using a Turrax for 5 min, 16000 rpm, and the suspension was spray-dried using a mini spray-dryer Buchi-191 with a 0.5 mm nozzle. The following parameters were adjusted in the spray drying process: spray feed rate of 0.01%, inlet air drying temperature of 180 °C, outlet air drying temperature of 104 °C, aspirator efficiency of 70%, and air pressure of 3.5 kgf cm$^{-2}$. For in vivo assay, all the protocols were approved by the ethics committee of the School of Pharmaceutical Sciences - Araraquara - SP - Brazil for animal care and use (CEUA/ FCFAr: 13/2021) and were conducted according to the principles of the National Council for Animal Experiments Control (CONCEA), based on the National Institutes of Health (NIH) Guidelines for the Care and Use of Laboratory Animals. We used 32 heterogeneous male rats, *Rattus norvegicus* (Wistar), at the age of 7 weeks. The induction of gastric ulceration in vivo was based on previously reported protocols[70–72] with modifications. The rats were randomly assigned to 4 groups: Positive control (PC), omeprazole (OME) 40 mg/Kg, AF-HA coacervate microparticles (MP) 33.3 mg/kg, and negative control (NC), with 8 animals per group. Omeprazole is a prodrug of the class of proton pump inhibitors that act by inhibiting the activity of H$^+$, K$^+$ ATPases, which are essential for acid production in the stomach, thus neutralizing acid secretion. Proton pump inhibitors have proven to be an effective treatment option in patients with gastrointestinal pathologies, thus reducing gastric ulcers induced by NSAIDs. In this way, the comparison of the gastric protective effect of the microparticles developed in this work with omeprazole in the in vivo model is relevant since omeprazole is a drug that is very commonly prescribed as a gastric protector[73,74].

They received, for seven consecutive days, orally (gavage), once a day, their respective treatments, except for the positive and negative controls, who received yogurt as a vehicle (Vigor© Grego, plain flavor, Vigor Alimentos S.A., Sao Paulo-SP, BRA). After the last day of treatment, the animals were kept without food for 18 h, and indomethacin 60 mg/kg was administered by gavage for all groups except the negative control. After 6 h the animals were euthanized. The stomach of the rats was carefully removed and opened along the greater curvature, and rinsed with saline. We used ImageJ®software to analyze the lesions. To determine the statistical significance, we used a one-way analysis of variance (ANOVA), followed by Fisher's LSD test, $p = 0.0003$.

### Reporting summary

Further information on research design is available in the Nature Portfolio Reporting Summary linked to this article.

## Data availability

The authors declare that all data supporting the findings of this study are available within the paper and its Supplementary Information files or available from the corresponding author upon request.

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

## Acknowledgements

The authors would like to thank T. Schwarz, A. G. Bittermann, and S. Handschin from the ScopeM for assisting with the optical confocal and the scanning electron microscopy. They acknowledge the support of H. Almohammadi and Y.Yuan from ETH Zurich for polarization and optical microscopy. The authors also acknowledge EMPA for the access to the X-Ray photoelectron spectroscopy instrumentation.

## Author contributions

M.P. and R.M. conceived the idea and designed the research. M.P. and S.K. carried out the coacervation experiments. M.P. characterized the coacervate and compiled the figures. J.Z. performed the AFM and AFM-IR and analyzed their data. V.L. performed the WAXS and analyzed the data. F.D.V., A.B.M and M.C. developed the microparticles for the in vivo study. F.D.V., L.S., T.M.B. and M.C. designed and performed the in vivo experiments. M.P. and R.M. wrote the manuscript. All authors edited and approved the final manuscript.

## Competing interests

The authors declare no competing interests.
