## [Peer Review File · Nature Communications]

Amyloid-polysaccharide interfacial coacervates as therapeutic materialsREVIEWER COMMENTS

Reviewer #1 (Remarks to the Author):

This is an interesting paper that describes the formation of novel materials using amyloid fibrils and polysaccharides that phase separate to generate interesting asymmetrical structures. These materials are then applied to demonstrate therapeutic effect.

In the introduction references 18-20 are all good examples from the current team using agrifood waste but there are some efforts in this area that predate this work, see Juliet Gerrards work on crystallins from fish eyes as an example

(https://scholar.google.ca/citations?view_op=view_citation&hl=en&user=q-niTPsAAAAJ&cstart=100&pagesize=100&citation_for_view=q-niTPsAAAAJ:WqliGbK-hY8C).

On page 5, the fibrils are described as being virtually infinite in length. This is not really accurate. Can an estimate of length be given in micrometers or another relevant scale.

On line 113 the thickness of dried spun fibres is given as 20 μm . Can an error be given to give a better representation of variability here and the number of samples examined?

Line 122 references the colour maps of the capsules. The images indicate that the colour is not homogeneous and that the film itself may be heterogeneous across the surface of the capsule. Can the authors comment? Is homogeneity needed for the applications proposed? Why might domains form and could these impact on the mechanical strength?

Figure 2d contains the phase diagram but it is somewhat difficult to read quickly. The figure has the actual data points plotted but it may be clearer to indicate the regions of the different phases with colouring of the sections of the phase diagram. This has been applied to phase diagrams in the SI.

Line 174 - is there a size limit to the capsules? Either a minimum size or a maximum size? Understanding this would allow a better evaluation of the potential of the technology. A quantitative description of their shape would also be interesting, as they appear to vary in the images in the SI and this may be somewhat of a disadvantage.

Line 185 the film is described as being homogeneous but Figure 2c indicates some heterogeneity at this length scale.

Line 186- the work is described as 'remarkable' but this would be better replaced with more balanced language.

Line 202: the amyloid fibrils are described as dominating the top surface of the film but also potentially diffusing to the bottom side too. This is surprising given the length scale of the fibrils. Can this tell us anything about the diffusion or mixing of these species? Might this be the same or different with other types of fibrils?

Line 201 describes the bottom surface of the film as being a random spongy morphology. Is this appearance consistent with the appearance of other HA based films? A comparison and reference to the literature would help the reader to understand here.

The chemical structures in Figure 3, part g, are low resolution and hard to see, can they be improved?

The microparticles used for the therapeutic demonstration have been described in another publication.

Are the spray dried particles characterized in these two studies? It would be helpful to describe this here in the current manuscript. What is not clear here is whether the rehydration properties and material

stability of the particles once introduced to liquid again have also been studied. Can the authors be sure that the particles have good integrity when they are injected for animal studies?

On line 270 the anti-inflammatory properties of b-lactoglobulin are described. Is this in the fibril form though?

At line 275 it would be good to make it clear that the capsules are being used in place of the drug control and that this is not a drug delivery application, which coincidentally would be interesting too. The conclusion states that that the system is easily tuneable but to what extent has this property been demonstrated? The capsules have been made but the discussion doesn't really describe tuning. The description of scope well beyond 'mere' therapeutics could also be more balanced by rephrasing and removing 'mere'. Further, the conclusion ends with reference to an example that is provided in the SI. This is not the usual form for a conclusion, i.e. it is not standard to introduce new data in a conclusion, particularly not in the last line. Can this data be introduced earlier in the manuscript and if worthwhile described in greater detail in the text?

In the methods the presence of fibrils is described via a qualitative assay but what about the quantification of the fibrils present? At what concentration are these species present?

At line 313, the films formed are described as being smeared i.e. one phase on top of another. Using what instrument and what method? Can the method be described adequately so that another scientist might repeat the method?

At line 355 there are two typos – 700 appears twice and the fibril stalk is described as a wire.

At line 372 italics are needed for *in vivo*.

Reviewer #2 (Remarks to the Author):

Peydayesh et al. Nat Chem, Amyloid-polysaccharide materials.

This study explores protein-polysaccharide coacervates for overcoming the chemical and mechanical instabilities of protein-based condensates. Their international collaboration demonstrates control over morphological diversity, from flexible sheets and fibers to capsules of coacervates, prepared from the polyanion polysaccharide hyaluronic acid and denaturation-induced amyloid fibers of the cationic β -lactoglobulin from whey. Cross- β protein assemblies are considered very stable, and irregular shapes apparent in the microparticle capsules may arise from these co-assembled fibers. Both IR and WAXS analyses support their presence in the resulting film composites.

The authors tested the biological stability by feeding rats the coacervate microparticles in gastric ulcer protection assays and demonstrated reduction in stomach lesion area. The authors should explain the robustness of this single assay strategy, even as a proof of principle experiment. The authors further speculate that both the protein and polysaccharide components have anti-inflammatory and antioxidant activity that may contribute to the observed protections, yet there is no evidence presented that the fibers are being broken down in the stomach to release functional protein. The relevance of the comparison to omeprazole, proven to reduce stomach acid, does not inform the mechanistic proposal

that is the foundation for their designed strategy.

Overall the paper is well written and opens exciting areas for new therapeutic coacervate strategies but covers a great deal of ground. From conceptual design and structural characterization to in vivo assays of mechanism of action, with each being given rather brief experimental and background coverage. For example: How critical is the polysaccharide for the observed biological effect – linear, branched, anionic? Can any cationic amyloid fiber be used to stabilize the coacervate? What is the structure of the coacervate co-assembly, random or ordered packing, and is that order important for stability and activity? How is the observed activity impacted by the nature of the initial wound?

Reviewer #3 (Remarks to the Author):

In this manuscript the authors report the formation of a range of materials formed from a polysaccharide (HA) and a protein that had been processed to form amyloid fibrils (AF). The authors demonstrate the ability to form films, fibers, and capsules via the simple mixing of the negatively charged HA and the positively charged AF. The authors characterize the formation of these materials as a function of mixing ratio (and initial biopolymer concentration) as well as the pH of the HA and AF solutions. The authors also analyze the chemical composition of the films by various spectroscopies (AFM-IR, XPS, WAXS) in an attempt to show that these materials are formed via coacervation of the polysaccharide and amyloid fibrils. Finally, the authors report a protective effect of microparticle capsules formed via this process on the formation of gastric ulcers.

Overall, the authors report a panel of materials with a range of interesting properties. However, I believe the manuscript would be strengthened through either the inclusion of more evidence to support the claims made herein (see examples below) or by significantly editing the text to rely only on the data presented.

For example, based on the reported data it is somewhat unclear that these materials are forming via complex coacervation, or any type of liquid-liquid phase separation at all. For example, the authors report several solid materials (films, fibers) and it is not clear that in fact a second liquid phase, at equilibrium with a dilute phase, is forming or rather if the two oppositely charged biopolymers are simply aggregating or precipitating (i.e. forming solid polyelectrolyte complexes). It is not clear from the reported methods exactly how the final characterized materials were prepared after their initial formation (were they dried? re-hydrated?). Again it is not clear if these materials are sensitive to the ionic strength of the solution (during or after formation), which would suggest that they are indeed formed by electrostatic interactions or if any molecular rearrangements (or equilibrations) occur in the presence of salt (e.g. are the asymmetries observed in the films present if the films are equilibrated with some salt). Finally, for the final biological characterization of the materials, while the authors report nice decreases in the lesions they also describe the mechanism for protection, which I did not see to be supported by any data.

Reviewers' comments:

Reviewer #1 (Remarks to the Author):

This is an interesting paper that describes the formation of novel materials using amyloid fibrils and polysaccharides that phase separate to generate interesting asymmetrical structures. These materials are then applied to demonstrate therapeutic effect.

We thank the reviewer for the positive comments on our manuscript; we have followed his/her suggestions to revise the manuscript.

In the introduction references 18-20 are all good examples from the current team using agrifood waste but there are some efforts in this area that predate this work, see Juliet Gerrards work on crystallins from fish eyes as an example (https://scholar.google.ca/citations?view_op=view_citation&hl=en&user=q-niTPsAAAAJ&cstart=100&pagesize=100&citation_for_view=q-niTPsAAAAJ:WqliGbK-hY8C).

We thank the reviewer for bringing our attention to this paper. We are familiar with Prof. Gerrard's works and included the mentioned example in the introduction, accordingly.

Furthermore, they can be obtained sustainably and cost-effectively by valorizing agri-food industry waste¹⁸, such as whey¹⁹, proteins from oilseed meals²⁰, and marine fisheries²¹.

²¹ Kaur, M. et al. Stability and cytotoxicity of crystallin amyloid nanofibrils. *Nanoscale* 6, 13169-13178, doi:10.1039/C4NR04624B (2014).

On page 5, the fibrils are described as being virtually infinite in length. This is not really accurate. Can an estimate of length be given in micrometers or another relevant scale.

We think there is a misunderstanding regarding the terminology of fibrils and fibers. Fibrils refer to amyloid fibrils, and fibers refer to the coacervate fiber we spun from the solution.

On page 5, the word "fibers" was used, and the discussion is about the continuous forming of interface string due to the coacervation of amyloid fibrils and hyaluronic acids.

Nevertheless, for the estimation of fibrils length, we have provided more information in the paper:

The presence of β -lactoglobulin AF was confirmed by birefringence using cross-polarized light and AFM (Figure S20). They have diameters and lengths ranging from 2 to 8 nm and 5-10 μm , respectively.

Figure S20. AFM image of β -lactoglobulin AF.

On line113 the thickness of dried spun fibres is given as 20 μm . Can an error be given to give a better representation of variability here and the number of samples examined?

We thank the reviewer for this comment. We have produced many fibers (more than 20) and evaluated their thicknesses by different methods (SEM, AFM, and light microscopy). The measured thickness varied between 18-22 μm , so a thickness value of around 20 μm was reported. Based on this comment, we removed "around" from the text and provided the value with the error as $20\pm 2 \mu\text{m}$.

The thickness of the resulting dry fibers was $20 \pm 2 \mu\text{m}$, as shown in the optical microscopy, atomic force microscopy (AFM), and scanning electron microscopy (SEM) images in **Figure S4**.

Line 122 references the colour maps of the capsules. The images indicate that the colour is not homogeneous and that the film itself may be heterogeneous across the surface of the capsule. Can the authors comment? Is homogeneity needed for the applications proposed? Why might domains form and could these impact on the mechanical strength?

The heterogeneity in colour maps comes from the PolScope cross-polarized system which indicates the nematic field, not inhomogeneities, and therefore shows only the change in the angle of fibrils. The surface of coacervate products is characterized as integrated and homogenous, and they possess the required mechanical stabilities for proposed applications.

Figure 2d contains the phase diagram but it is somewhat difficult to read quickly. The figure has the actual data points plotted but it may be clearer to indicate the regions of the different phases with colouring of the sections of the phase diagram. This has been applied to phase diagrams in the SI.

We thank the reviewer for this comment. We have revised Figure 2, Figure S8 and S10, accordingly.

Figure 2. Photographs of **a)** films, **b)** fibers and **c)** capsules formed by coacervation of AF and HA (The colorful image in panel c shows the direction of the fibrils in the plane, see the colormaps in the top-right corner). **d)** Concentration phase diagram for producing capsules (pH 3), AF as the inner phase and HA as the outer phase and **e)** HA as the inner phase and AF as the outer phase. **f)** pH phase diagram for producing capsules (AF 2wt.% and HA 1wt.%). AF as the inner phase and HA as the outer phase and **g)** HA as the inner phase and AF as the outer phase. Confocal microscopy of **h)** rhodamine-labeled AF, **i)** the fluorescein-labeled HA within the capsule wall, **j)** merged image (confirming colocalization of the AF and HA within the capsule's wall) and **k)** transmission image.

Figure S8. Concentration phase diagram for producing fibers (AF as the inner phase and HA as the outer phase).

Figure S10. Concentration phase diagram for producing films (AF as the top phase and HA as the bottom phase).

Line 174 -is there a size limit to the capsules? Either a minimum size of a maximum size?

Understanding this would allow a better evaluation of the potential of the technology. A

quantitative description of their shape would also be interesting, as they appear to vary in the images in the SI and this may be somewhat of a disadvantage.

There is no size limit for capsule production, and the injected volume (inner phase) determines the size of the capsules. This can be from micron size, for instance, using a microfluidic device, to the millimeter and macroscopic size using a needle.

Line 185 the film is described as being homogeneous but Figure 2c indicates some heterogeneity at this length scale.

Figure 2c shows a colorful image map of the capsule indicating the nematic field and not the film. As explained in previous comments, the heterogeneity in colour maps comes from the PolScope bending and shows only the change in the angle of fibrils. The surface of coacervate products is characterized as integrated and homogenous, and they possess the required mechanical stabilities for all the proposed applications.

Line 186- the work is described as 'remarkable' but this would be better replaced with more balanced language.

We agree with the reviewer, and the word "remarkably" was changed to "Furthermore".

Furthermore, infrared spectra were collected at the indicated location on the AFM images **(Figure 3c)**.

Line 202: the amyloid fibrils are described as dominating the top surface of the film but also potentially diffusing to the bottom side too. This is surprising given the length scale of the fibrils. Can this tell us anything about the diffusion or mixing of these species? Might this be the same or different with other types of fibrils?

We thank the reviewer for this comment. We were also first surprised by this result, but three independent techniques, i.e., SEM, AFM-IR, and XPS, confirmed the diffusion of amyloid

fibrils to the other side of the coacervate layer. The low diffusion of HA can be related to its higher viscosity, hindering its transfer to the other side of the layer.

Line 201 describes the bottom surface of the film as being a random spongy morphology. Is this appearance consistent with the appearance of other HA based films? A comparison and reference to the literature would help the reader to understand here.

Yes, this random spongy morphology was also observed in other previously published works on HA-based coacervate films, for example, in *Hyaluronic Acid/Chitosan Coacervate-Based Scaffolds*, *Biomacromolecules* 2018, 19, 4, 1198–1211. We have updated the manuscript based on this comment.

Figure 3f, the bottom surface of the coacervate film has a more random spongy morphology (similar to other reports on HA-based coacervate films⁴⁶), yet some AF can be seen on this side (see inset).

⁴⁶ Karabiyik Acar, O., Kayitmazer, A. B. & Torun Kose, G. *Hyaluronic Acid/Chitosan Coacervate-Based Scaffolds*. *Biomacromolecules* 19, 1198-1211, doi:10.1021/acs.biomac.8b00047 (2018).

The chemical structures in Figure 3, part g, are low resolution and hard to see, can they be improved?

Yes, we have increased the resolution of chemical structures in Figure 3, accordingly.

Furthermore, some problems with the resolution of figures are generated during the conversion of the .doc file to pdf; thus, for final publication purposes, the figures in .pdf format will be provided to the journal.

Figure 3. Characterization of AF-HA coacervate films at different sides and interfaces. **a)** The scheme of AFM-IR experimental setup while scanning on the film. **b)** Morphological images (left), maps of infrared (IR) absorption in the Amide I at 1654 cm^{-1} (middle) and stiffness that indicates the nanomechanical properties of the top (upper) and the bottom (lower) side of the film. The crosses refer to the locations for performing IR spectroscopy measurements. The scale bar is 500 nm . **c)** Nanoscale localized infrared spectrum of top and bottom side of the film collected at the crossed location in panel b. SEM images **d)** surface of AF side, **e)** AF-HA interface and **f)** surface of HA side. **g)** XPS C1s and O1s chemical state spectra of pure AF and HA, as well as top and bottom surfaces of coacervate films. 2D WAXS scattering patterns of **h)** pure AF and **i)** AF+HA films. **j)** Intensity I as a function of the azimuthal angle (χ) of AF and AF+HA films. **k)** Intensity I as a function of scattering vector q of AF and AF+HA films, as well as HA powder.

The microparticles used for the therapeutic demonstration have been described in another publication. Are the spray dried particles characterized in these two studies? It would be helpful to describe this here in the current manuscript. What is not clear here is whether the rehydration properties and material stability of the particles once introduced to liquid again have also been studied. Can the authors be sure that the particles have good integrity when they are injected for animal studies?

This might be another misunderstanding since we have not previously reported any work on AF-HA coacervate microparticles.

The microparticles developed were suspended (and not solubilized) in yogurt immediately before use in the *in vivo* assay, only to facilitate the administration of this multiparticulate system. No stock dispersions were prepared for storage. Although no stability tests were carried out, the group's experience in this type of system, supported by daily practice and the literature, together with the extemporaneous preparation of the dispersions, are indicative of the non-solubilization of the microparticles. According to the literature¹, HA is an acidic polymer with carboxylic groups and presents an acid dissociation constant (pKa) close to 4. For this reason, when it is found in media with an acid pH, as is the case of yogurt, the polymer chains are strongly interlinked so that only an increase in pH close to neutrality is able to ionize the carboxylic groups, promoting the repulsion of the polymeric network for the consequent neutralization of intermolecular forces, which configures a behavior of pH-dependent solubility. It should also be noted that the polyelectrolyte complexation promoted between the opposite charges of HA and AF prior to drying by spray drying contributes strongly to maintaining the system's structure in vehicles for oral administration, prepared extemporaneously.

1 Mero, A. & Campisi, M. Hyaluronic Acid Bioconjugates for the Delivery of Bioactive Molecules. *Polymers* 6, 346-369 (2014).

On line 270 the anti-inflammatory properties of β -lactoglobulin are described. Is this in the fibril form though?

As reported in the paper, β -Lactoglobulin possesses anti-inflammatory and peripheral antinociceptive activities⁶². It is a rich source of sulfur amino acids such as cysteine that can stimulate the synthesis of glutathione, one of the compounds responsible for the anti-inflammatory and antioxidant activity in various tissues and organs⁶³.

The reported anti-inflammatory properties of β -Lactoglobulin in Kathem et al. work⁶², were obtained by oral administration in the monomer form. Once proteins reach the stomach, enzymes and hydrochloric acid break them down into smaller chains of amino acids. The resulting amino acids, after digestion of β -Lactoglobulin, such as glycine, cysteine, and glutamic acid, can form glutathione, which is an active compound in tissue building and repair. To show that the β -Lactoglobulin AF are digestible and similar anti-inflammatory effects can be expected from them, we performed the *in vitro* digestion of AF-HA capsules containing AF as an inner phase. The digestion was done at 37 °C for 2 h using fresh simulated gastric fluid (SGF) containing pepsin (as a control, the same experiment was done with Milli-Q water). As observed in the control sample in **Figure S19**, after 2 h incubation at 37 °C, the long AF were detected in the water by AFM, confirming the presence and release of AF from the coacervate capsule to the bulk of water. However, for the sample treated with SGF, the AFM image did not show any evidence of the existence of AF but short fragments, indicating the complete digestion of AF. As HA has adhesive properties⁶⁴, the hypothesis that we suggest for the protective effect of microparticles is that they can adhere to the mucosa, and then HA and AF may synergistically contribute to the ulcer gastric protective properties of the coacervate microparticles, highlighting them as systems with efficient biological activity for gastric ulcer protection and possible therapeutic use.

Figure S19. *in vitro* digestion of AF released from AF-HA capsules.

For the method:

***in vitro* digestion of AF-HA coacervate**

The *in vitro* digestion of AF-HA coacervate was applied to simulate the physiological gastric digestion process by following the INFOGEST standard protocol⁶⁷. Briefly, fresh SGF containing the pepsin from porcine gastric mucosa (P6887) with the enzyme activity of 500 U/mL was prepared and equilibrated to 37 °C before the digestion test. One AF-HA coacervate capsule was digested in 3 mL of SGF solution, followed by the addition of CaCl₂ solution (0.3 M) into the digestion solution reaching a final concentration of 0.15 mM prior to digestion. The digestion was carried out in a water bath (VWR 462-0493) at 37 °C under shaking conditions (60 rpm) for 120 min, and AFM imaging was performed immediately after digestion. The control experiment was performed using Milli-Q water instead of an SGF solution.

62 Kathem, R. A. et al. Antinociceptive and Anti-Inflammatory Activity of Bovine Milk-Derived β -Lactoglobulin: Study of Combined Effect with Tramadol or Indomethacin. *Global Journal of Pharmacology* 8, 444-456, doi:10.5829/idosi.gjp.2014.8.3.1119 (2014).

63 Rosaneli, C. F., Bighetti, A. E., Antônio, M. A., Carvalho, J. E. & Sgarbieri, V. C. Protective Effect of Bovine Milk Whey Protein Concentrate on the Ulcerative Lesions Caused by Subcutaneous Administration of Indomethacin. *Journal of Medicinal Food* 7, 309-314, doi:10.1089/jmf.2004.7.309 (2004).

64 Marinho, A., Nunes, C. & Reis, S. Hyaluronic Acid: A Key Ingredient in the Therapy of Inflammation. *Biomolecules* 11, doi:10.3390/biom11101518 (2021).

The above yellow-highlighted information has been added to the manuscript accordingly.

At line 275 it would be good to make it clear that the capsules are being used in place of the drug control and that this is not a drug delivery application, which coincidentally would be interesting too.

We thank the reviewer for the comment and suggestion. In this project, we used microparticles composed of HA and AF as a drug with gastric ulcer protective properties due to the anti-inflammatory properties of these compounds. However, we did some preliminary work on the drug encapsulation, and it was possible to incorporate curcumin into microparticles (please refer to Figures S6 and S13). To do so, first, we dissolved curcumin in AF, and then the coacervation with HA was performed. This showed that the microparticles developed in this project not only have therapeutic activity but are also capable of delivering drugs. In this short communication, we focused on the use of HA and AF, but the application of the microparticles developed here as a drug delivery system will be investigated in depth in further studies.

The conclusion states that that the system is easily tuneable but to what extent has this property been demonstrated? The capsules have been made but the discussion doesn't really describe tuning. The description of scope well beyond 'mere' therapeutics could also be more balanced by rephrasing and removing 'mere'. Further, the conclusion ends with reference to an example that is provided in the SI. This is not the usual form for a conclusion, i.e. it is not standard to introduce new data in a conclusion, particularly not in the last line. Can this data be introduced earlier in the manuscript and if worthwhile described in greater detail in the text?

We thank the reviewer for his/her valuable comment. The tunability of coacervates was presented by engineering the coacervation phenomenon to produce various products, including

films, capsules, fibers, and microparticles. Moreover, the size of coacervates is tunable by varying the injected volumes. Additionally, the change in strength and mechanical stability of coacervates was shown as a function of pH and initial precursor concentrations.

We agree with the reviewer on removing "mere", and this word is deleted from the conclusion accordingly.

... we believe that these materials may have a scope well beyond the therapeutics...

Regarding ending the conclusion by referring to the water purification application, we totally agree with the reviewer, and it is not standard to finish a paper like that; however, we did not want to distract the focus from the main application of the prepared coacervate as therapeutic materials. To fix this issue, we have removed this sentence from the conclusion, and based on the reviewer's suggestion, we mentioned it earlier and only at the end paragraph of the introduction.

In addition, to show the generality of the approach, we present in the SI one further application example of coacervates beyond therapeutics, specifically designed for water purification applications.

In the methods the presence of fibrils is described via a qualitative assay but what about the quantification of the fibrils present? At what concentration are these species present?

Per the reviewer's request, we have performed AFM to quantify the amyloid fibrils:

β -lactoglobulin was isolated from whey according to the previous protocols⁶⁵. Then, AF were formed by fibrillation of purified β -lactoglobulin (2-4 wt.%) in an aqueous medium at 90 °C and a pH of 2. The conversion rate from monomer to AF for β -lactoglobulin is around 80%⁶⁶. More information on the preparation of β -lactoglobulin AF can be found in our previous paper⁶⁶. The presence of β -lactoglobulin AF was confirmed by birefringence using cross-

polarized light and AFM (**Figure S20**). They have diameters and lengths ranging from 2 to 8 nm and 5-10 μm , respectively.

Figure S20. AFM image of β -lactoglobulin AF.

65 Viamontes, J. & Tang, J. X. Continuous isotropic-nematic liquid crystalline transition of F-actin solutions. *Physical Review E* **67**, 040701, doi:10.1103/PhysRevE.67.040701 (2003).

66 Jung, J.-M., Savin, G., Pouzot, M., Schmitt, C. & Mezzenga, R. Structure of Heat-Induced β -Lactoglobulin Aggregates and their Complexes with Sodium-Dodecyl Sulfate. *Biomacromolecules* **9**, 2477-2486, doi:10.1021/bm800502j (2008).

At line 313, the films formed are described as being smeared i.e. one phase on top of another. Using what instrument and what method? Can the method be described adequately so that another scientist might repeat the method?

The films were prepared simply by placing one phase on top of the other without using any instrument or particular methods.

The details are already available in the manuscript as follows:

When the AF solution was uniformly layered on top of the HA solution (in an equal volume ratio), coacervation occurred, and a membrane film formed immediately at the interface between the two solutions (**Figure 1a**). The order in which the two solutions were combined (HA over the AF) could be changed; however, due to the higher viscosity of the HA, uniformly

spreading it on top of the AF solution to achieve a homogeneous membrane film was more difficult.

At line 355 there are two typos – 700 appears twice and the fibril stalk is described as a wire.

We thank the reviewer for this comment and have corrected the manuscript accordingly.

At line 372 italics are needed for *in vivo*.

We thank the reviewer for this comment and have corrected the manuscript accordingly.

Reviewer #2 (Remarks to the Author):

Peydayesh et al. Nat Chem, Amyloid-polysaccharide materials.

This study explores protein-polysaccharide coacervates for overcoming the chemical and mechanical instabilities of protein-based condensates. Their international collaboration demonstrates control over morphological diversity, from flexible sheets and fibers to capsules of coacervates, prepared from the polyanion polysaccharide hyaluronic acid and denaturation-induced amyloid fibers of the cationic β -lactoglobulin from whey. Cross- β protein assemblies are considered very stable, and irregular shapes apparent in the microparticle capsules may arise from these co-assembled fibers. Both IR and WAXS analyses support their presence in the resulting film composites.

The authors tested the biological stability by feeding rats the coacervate microparticles in gastric ulcer protection assays and demonstrated reduction in stomach lesion area. The authors should explain the robustness of this single assay strategy, even as a proof of principle experiment.

We thank the reviewer for the comment, and the below yellow-highlighted text has been added to the manuscript to explain the robustness of this single assay strategy.

Ulcers are caused by a gastrointestinal disorder that occurs when there is an imbalance between aggressive factors (secretion of HCl and pepsin) and factors that defend the gastric mucosa, such as secretion of bicarbonate and mucus, and production of prostaglandins, causing superficial damage that is generated by an inflammatory process that promotes tissue loss on the mucosal surface. HCl and pepsin are the main aggressors of the gastric mucosa, and their damage can be minimized or avoided by the protective factors of the gastric mucosa, such as the presence of an intact layer of gastric epithelial cells, a protective layer of mucin, secretion of bicarbonate that neutralizes the acidity present in the gastric mucosa, presence of growth factors, angiogenic factors and the presence of prostaglandins⁵⁰. Prostaglandins decrease gastric

secretion, increase mucosal blood flow and regulate gastric mucin synthesis, stimulate mucus and bicarbonate secretion, induction of angiogenesis, and various other functions to restore mucosal integrity, playing a pivotal role in ulcer healing process⁵¹. Studies on this disease are extremely relevant since it is considered the most common chronic disease of the upper digestive tract in adults, affecting about 4 million people per year worldwide⁵². There are several etiological factors linked to the cause of these diseases, such as smoking, stress, nutritional deficiencies, abusive and prolonged use of drugs such as non-steroidal anti-inflammatory drugs (NSAIDs), alcohol, and *H. pylori* infection⁵³. However, the most common factor is associated with the intake of NSAIDs, which are among the most used drugs in the world^{54,55}. In this sense, this protocol using indomethacin, an NSAID to induce gastric ulcers, is important in the study of this disease.

Indomethacin-related gastrointestinal toxicity occurs by inhibiting the pathway of cyclooxygenase 1 and 2 (COX-1 and COX-2) enzymes, which are responsible for the synthesis of prostaglandins. In addition, after the formation of the lesion in the mucosa, NSAIDs inhibit the first restitution events necessary to repair the initial superficial damage, as well as subsequent events, acting in the reduction of epithelial cell proliferation and in the reduction of angiogenesis, promoting a delay in the ulcer healing⁵⁶.

As HA has adhesive properties⁶⁴, the hypothesis that we suggest for the protective effect of microparticles is that they can adhere to the mucosa, and then HA and AF may synergistically contribute to the ulcer gastric protective properties of the coacervate microparticles, highlighting them as systems with efficient biological activity for gastric ulcer protection and possible therapeutic use.

It is still worth mentioning that the scope of this manuscript is to produce amyloid-polysaccharide coacervate and demonstrate several applications, among them the therapeutic application; of course, the optimization of the therapeutic activity and detailed action

mechanism are beyond the scope of this first communication and should be addressed in a follow-up study.

50 Srikanta, B. M., Sathisha, U. V. & Dharmesh, S. M. Alterations of matrix metalloproteinases, gastric mucin and prostaglandin E2 levels by pectic polysaccharide of swallow root (*Decalepis hamiltonii*) during ulcer healing. *Biochimie* 92, 194-203, doi:<https://doi.org/10.1016/j.biochi.2009.10.005> (2010).

51 Arakawa, T. et al. Quality of ulcer healing in gastrointestinal tract: its pathophysiology and clinical relevance. *World J Gastroenterol* 18, 4811-4822, doi:[10.3748/wjg.v18.i35.4811](https://doi.org/10.3748/wjg.v18.i35.4811) (2012).

52 Abbasi-Kangevari, M. et al. Quality of care of peptic ulcer disease worldwide: A systematic analysis for the global burden of disease study 1990-2019. *PLoS One* 17, e0271284, doi:[10.1371/journal.pone.0271284](https://doi.org/10.1371/journal.pone.0271284) (2022).

53 Thorsen, K., Søreide, J. A., Kvaløy, J. T., Glomsaker, T. & Søreide, K. Epidemiology of perforated peptic ulcer: age- and gender-adjusted analysis of incidence and mortality. *World J Gastroenterol* 19, 347-354, doi:[10.3748/wjg.v19.i3.347](https://doi.org/10.3748/wjg.v19.i3.347) (2013).

54 Sostres, C., Gargallo, C. J., Arroyo, M. T. & Lanas, A. Adverse effects of non-steroidal anti-inflammatory drugs (NSAIDs, aspirin and coxibs) on upper gastrointestinal tract. *Best Practice & Research Clinical Gastroenterology* 24, 121-132, doi:<https://doi.org/10.1016/j.bpg.2009.11.005> (2010).

55 Narayanan, M., Reddy, K. M. & Marsicano, E. Peptic Ulcer Disease and *Helicobacter pylori* infection. *Mo Med* 115, 219-224 (2018).

56 Musumba, C., Pritchard, D. M. & Pirmohamed, M. Review article: cellular and molecular mechanisms of NSAID-induced peptic ulcers. *Alimentary Pharmacology & Therapeutics* 30, 517-531, doi:<https://doi.org/10.1111/j.1365-2036.2009.04086.x> (2009).

64 Marinho, A., Nunes, C. & Reis, S. Hyaluronic Acid: A Key Ingredient in the Therapy of Inflammation. *Biomolecules* 11, doi:[10.3390/biom11101518](https://doi.org/10.3390/biom11101518) (2021).

The authors further speculate that both the protein and polysaccharide components have anti-inflammatory and antioxidant activity that may contribute to the observed protections, yet there is no evidence presented that the fibers are being broken down in the stomach to release functional protein.

As reported in the paper, β -Lactoglobulin possesses anti-inflammatory and peripheral antinociceptive activities⁶². It is a rich source of sulfur amino acids such as cysteine that can stimulate the synthesis of glutathione in various tissues and organs⁶³.

The reported anti-inflammatory properties of β -Lactoglobulin in Kathem et al. work⁶², were obtained by oral administration in the monomer form. Once proteins reach the stomach, enzymes and hydrochloric acid break them down into smaller chains of amino acids. The

resulting amino acids, after digestion of β -Lactoglobulin, such as glycine, cysteine, and glutamic acid, can form glutathione, which is an active compound in tissue building and repair. In order to substantiate this point, the following text and experiments were added to the manuscript:

To show that the β -Lactoglobulin AF are digestible and similar anti-inflammatory effects can be expected from them as for the monomers, we performed the *in vitro* digestion of AF-HA capsules containing AF as an inner phase. The digestion was done at 37 °C for 2 h using fresh simulated gastric fluid (SGF) containing pepsin (as a control, the same experiment was done with Milli-Q water). As observed in the control sample in **Figure S19**, after 2 h incubation at 37 °C, the long AF were detected in the water by AFM, confirming the presence and release of AF from the coacervate capsule to the bulk of water. However, for the sample treated with SGF, the AFM image did not show any evidence of the existence of AF but short fragments, indicating the complete digestion of AF. As HA has adhesive properties⁶⁴, the hypothesis that we suggest for the protective effect of microparticles is that they can adhere to the mucosa, and then HA and AF may synergistically contribute to the ulcer gastric protective properties of the coacervate microparticles, highlighting them as systems with efficient biological activity for gastric ulcer protection and possible therapeutic use.

Figure S19. *in vitro* digestion of AF released from AF-HA capsules.

For the method:

***in vitro* digestion of AF-HA coacervate**

The *in vitro* digestion of AF-HA coacervate was applied to simulate the physiological gastric digestion process by following the INFOGEST standard protocol⁶⁷. Briefly, fresh SGF containing the pepsin from porcine gastric mucosa (P6887) with the enzyme activity of 500 U/mL was prepared and equilibrated to 37 °C before the digestion test. One AF-HA coacervate capsule was digested in 3 mL of SGF solution, followed by the addition of CaCl₂ solution (0.3 M) into the digestion solution reaching a final concentration of 0.15 mM prior to digestion. The digestion was carried out in a water bath (VWR 462-0493) at 37 °C under shaking conditions (60 rpm) for 120 min, and AFM imaging was performed immediately after digestion. The control experiment was performed using Milli-Q water instead of an SGF solution.

62 Kathem, R. A. et al. Antinociceptive and Anti-Inflammatory Activity of Bovine Milk-Derived β -Lactoglobulin: Study of Combined Effect with Tramadol or Indomethacin. *Global Journal of Pharmacology* 8, 444-456, doi:10.5829/idosi.gjp.2014.8.3.1119 (2014).

63 Rosanelli, C. F., Bighetti, A. E., Antônio, M. A., Carvalho, J. E. & Sgarbieri, V. C. Protective Effect of Bovine Milk Whey Protein Concentrate on the Ulcerative Lesions Caused by Subcutaneous Administration of Indomethacin. *Journal of Medicinal Food* 7, 309-314, doi:10.1089/jmf.2004.7.309 (2004).

64 Marinho, A., Nunes, C. & Reis, S. Hyaluronic Acid: A Key Ingredient in the Therapy of Inflammation. *Biomolecules* 11, doi:10.3390/biom11101518 (2021).

67 Brodkorb, A. et al. INFOGEST static *in vitro* simulation of gastrointestinal food digestion. *Nature Protocols* 14, 991-1014, doi:10.1038/s41596-018-0119-1 (2019).

The relevance of the comparison to omeprazole, proven to reduce stomach acid, does not inform the mechanistic proposal that is the foundation for their designed strategy.

We thank the reviewer for the comment and provided more explanation in this regard as follows:

Omeprazole is a prodrug of the class of proton pump inhibitors that act by inhibiting the activity of H⁺, K⁺ ATPases, which are essential for acid production in the stomach, thus neutralizing acid secretion. Proton pump inhibitors have proven to be an effective treatment option in patients with gastrointestinal pathologies, thus reducing gastric ulcers induced by NSAIDs. In

this way, the comparison of the gastric protective effect of the microparticles developed in this work with omeprazole in the *in vivo* model is relevant since omeprazole is a drug that is very commonly prescribed as a gastric protector^{71,72}.

71 Lim, Y. J., Phan, T. M., Dial, E. J., Graham, D. Y. & Lichtenberger, L. M. *In vitro* and *in vivo* protection against indomethacin-induced small intestinal injury by proton pump inhibitors, acid pump antagonists, or indomethacin-phosphatidylcholine. *Digestion* **86**, 171-177, doi:10.1159/000339882 (2012).

72 Yanaka, A. Role of Sulforaphane in Protection of Gastrointestinal Tract Against *H. pylori* and NSAID-Induced Oxidative Stress. *Curr Pharm Des* **23**, 4066-4075, doi:10.2174/1381612823666170207103943 (2017).

Overall the paper is well written and opens exciting areas for new therapeutic coacervate strategies but covers a great deal of ground. From conceptual design and structural characterization to *in vivo* assays of mechanism of action, with each being given rather brief experimental and background coverage. For example: How critical is the polysaccharide for the observed biological effect – linear, branched, anionic? Can any cationic amyloid fiber be used to stabilize the coacervate?

The coacervation system here is based on electrostatic interaction. AF at pH below 5.2 are positively charged (cationic), and HA is an anionic negatively charged polysaccharide. Hence the best coacervation results are achieved at pH 3 with cationic amyloid fibrils.

Regarding the type of polysaccharides, we have pre-screening many anionic polysaccharides (i.e., agarose, dextran, gellan gum, xanthan gum, and pectin) for coacervation with cationic AF, and in the end, the coacervates with HA as a linear polysaccharide with a high molecular weight were the most robust and uniform ones.

It is possible, of course, to initiate the coacervation with anionic AF (at pHs above 5.2) and cationic polysaccharides such as chitosan (CS). To show this possibility, we ran the coacervation with these two precursors at pH 8. As observed in **Figure S17**, it is possible to produce AF sacs in the CS phase or CS sacs in the AF phase, as well as an interfacial film after layering them on top of each other.

Figure S17. Coacervation of anionic AF (2 wt.%) and cationic CS (2 wt.%) at pH 8.

The yellow-highlighted text has been added to the manuscript accordingly.

What is the structure of the coacervate co-assembly, random or ordered packing, and is that order important for stability and activity?

In this paper, we evaluated the structure of coacervates in detail by using different independent methods, including SEM, XPS, AFM-IR, and WAXS. The structure is asymmetric, and there is a gradient of amyloid fibrils concentration through the coacervate layer. One side of the coacervated layer is covered by amyloid fibrils, and the other side has a random spongy structure, yet some AF can be seen. As reported in the paper, the coacervate products, including capsules, films, fibers, and microparticles, exhibited high mechanical stability and robustness. Having amyloid fibrils with excellent activity in the structure, we observed high activity, as reflected by two presented applications for therapeutic materials and water purification.

How is the observed activity impacted by the nature of the initial wound?

Indomethacin acts by inhibiting the protective action of prostaglandins by inhibiting COX 1 and 2, which are involved in mucus production. This increases the intestinal epithelium's permeability and exposes the mucosa to aggressive agents such as bile and free radicals. Thus, the microparticles developed in this study demonstrate that they protect the mucosa from these agents and prevent the formation or reduction of gastric ulcers.

1 Bi, W. P., Man, H. B. & Man, M. Q. Efficacy and safety of herbal medicines in treating gastric ulcer: a review. World J Gastroenterol 20, 17020-17028, doi:10.3748/wjg.v20.i45.17020 (2014).

Reviewer #3 (Remarks to the Author):

In this manuscript the authors report the formation of a range of materials formed from a polysaccharide (HA) and a protein that had been processed to form amyloid fibrils (AF). The authors demonstrate the ability to form films, fibers, and capsules via the simple mixing of the negatively charged HA and the positively charged AF. The authors characterize the formation of these materials as a function of mixing ratio (and initial biopolymer concentration) as well as the pH of the HA and AF solutions. The authors also analyze the chemical composition of the films by various spectroscopies (AFM-IR, XPS, WAXS) in an attempt to show that these materials are formed via coacervation of the polysaccharide and amyloid fibrils. Finally, the authors report a protective effect of microparticle capsules formed via this process on the formation of gastric ulcers. Overall, the authors report a panel of materials with a range of interesting properties. However, I believe the manuscript would be strengthened through either the inclusion of more evidence to support the claims made herein (see examples below) or by significantly editing the text to rely only on the data presented.

We thank the reviewer for the positive comments on our manuscript; we have followed his/her suggestions to revise the manuscript.

For example, based on the reported data it is somewhat unclear that these materials are forming via complex coacervation, or any type of liquid-liquid phase separation at all. For example, the authors report several solid materials (films, fibers) and it is not clear that in fact a second liquid phase, at equilibrium with a dilute phase, is forming or rather if the two oppositely charged biopolymers are simply aggregating or precipitating (i.e. forming solid polyelectrolyte complexes). It is not clear from the reported methods exactly how the final characterized materials were prepared after their initial formation (were they dried? re-hydrated?). Again it is not clear if these materials are sensitive to the ionic strength of the solution (during or after formation), which would suggest that they are indeed formed by electrostatic interactions or if

any molecular rearrangements (or equilibrations) occur in the presence of salt (e.g. are the asymmetries observed in the films present if the films are equilibrated with some salt).

We thank the reviewer for the constructive comments on complex coacervation. We have performed some further experiments to clearly prove that the reported materials are formed by complex coacervation; more precisely, we have run inverted charges complex coacervation (chitosan and negatively charged AF), pH-dependent coacervation experiments and ionic-strength dependent experiments. The first experiment (chitosan-AF) has already been discussed in response to reviewer 2; here below we discuss the pH-dependent and ionic strength-dependent coacervation:

Coacervation is an associative liquid-liquid phase separation (LLPS) phenomenon that occurs through electrostatic interaction between oppositely charged macromolecular building blocks, such as proteins, polysaccharides and polymers in solution, resulting in a colloidal rich phase, the coacervate, and a remaining biomacromolecule-depleted phase in equilibrium with the denser coacervate^{22,23}.

During AF and HA coacervation, the second liquid phase, a dilute phase, is formed.

Figure S2, exhibits the bulk coacervation of AF and HA, where the coacervate and dilute phase (equilibrium solution) are clearly distinguishable. The equilibrium solution was transparent and had a very low viscosity, different from the two precursors.

Figure S2. Bulk coacervation of AF (2 wt.%) and HA (1 wt.%) at pH 3. The two resulting phases after coacervation, i.e., coacervate and diluted phases, are visible.

The electrostatic interaction is the driving force of complex coacervation. As such, a critical factor in the coacervation of AF and HA is pH, which directly determines the charge density of polyions and, thus, the electrostatic forces²⁵. **Figure S12** depicts the electrophoretic mobility of HA and AF over a wide pH range. Although HA has a net negative charge throughout the pH range, AF have an isoelectric point (IEP) of 5.2 and a positive charge at pHs below this value. Although the effect of pH on coacervation and pH phase diagrams were already presented (Figures 2f and g and Figures S13 and S14), to clearly reveal that the LLPS occurred due to electrostatic interactions (complex coacervation), we performed the bulk coacervation of AF and HA at pHs 9 (both AF and HA are in negatively charged) and 3 (AF has a positive charge and HA has a negative charge). As observed in **Figure S15** at pH 9 the AF-HA mixture is miscible, translucent and no gel was formed, but by decreasing the pH to 3 the coacervation between oppositely charged precursors occurred and resulted in robust, turbid gel, highlighting the predominate role of electrostatic interaction in this phenomenon.

Figure S15. The role of electrostatic interaction in bulk coacervation of AF (2 wt.%) and HA (1 wt.%).

For the characterization, the dried films were used in AFM, WAXS and XPS. For SEM, the structure of capsules was fixed by a solution containing 2% glutaraldehyde, 3% sucrose, and phosphate buffer at pH 7.4. Then the samples were dehydrated by solvent exchange against increasing concentrations of ethanol. Finally, the ethanol was removed by critical point drying to preserve the internal structure of the coacervate layer.

Another proof of complex coacervation (electrostatic interaction) is the sensitivity to the solution ionic strength. To evaluate the ionic strength effect, the bulk coacervation of AF and HA was done in the presence of salt (NaCl) at different concentrations, i.e., 0, 10, 100, and 1000 mM. **Figure S16** shows the visual appearance of coacervation samples as well as their turbidity as a function of salt concentration. As observed, strong coacervation occurred in the absence of salt, resulting in a coacervate phase and a transparent equilibrium solution. The turbidity of the coacervate phase was around 2000 a.u. By increasing the salt concentration, charge screening of precursors takes place, and coacervation becomes weaker. At the salt concentration of 1000 mM, the sample gel properties were lost, and homogenous and cloudy solutions with turbidity around 500 a.u were obtained.

Figure S16. The ionic strength effect on coacervation of AF (2 wt.%) and HA (1 wt.%) at pH 3. a) The visual appearance of coacervation samples in different salt concentrations. b) Turbidity of samples as a function of salt concentration.

22 Sing, C. E. & Perry, S. L. Recent progress in the science of complex coacervation. *Soft Matter* 16, 2885-2914, doi:10.1039/D0SM00001A (2020).

23 Eghbal, N. & Choudhary, R. Complex coacervation: Encapsulation and controlled release of active agents in food systems. *LWT* 90, 254-264.

25 Weinbreck, F., de Vries, R., Schrooyen, P. & de Kruif, C. G. Complex Coacervation of Whey Proteins and Gum Arabic. *Biomacromolecules* 4, 293-303, doi:10.1021/bm025667n (2003).

Finally, for the final biological characterization of the materials, while the authors report nice decreases in the lesions they also describe the mechanism for protection, which I did not see to be supported by any data

We have further elaborated this point with additional references and an ad-hoc run experiment to substantiate the mechanism:

The main mechanisms for gastric ulcer protection by AF-HA microparticles are schematically depicted in **Figure 4e**. The main protective effect is attributed to the presence of both HA and AF. HA presents antioxidant properties and is involved in several physiological processes, including wound repair and regeneration, morphogenesis, and matrix organization⁵⁹. The antioxidant property of HA is related to its hydroxyl functional groups that can absorb reactive oxygen species and avoid gastric mucosal damage. Additionally, HA can promote cell proliferation, increase growth factor, and stimulate fibroblasts to produce collagen fibre contributing to wound healing^{60,61}. Moreover, β -Lactoglobulin possesses anti-inflammatory and peripheral antinociceptive activities⁶². It is a rich source of sulfur amino acids such as cysteine that can stimulate the synthesis of glutathione, one of the compounds responsible for the anti-inflammatory and antioxidant activity in various tissues and organs⁶³. To show that the β -Lactoglobulin AF are digestible and similar anti-inflammatory effects can be expected from them, we performed the *in vitro* digestion of AF-HA capsules containing AF as an inner phase. The digestion was done at 37 °C for 2 h using fresh simulated gastric fluid (SGF) containing pepsin (as a control, the same experiment was done with Milli-Q water). As observed in the control sample in **Figure S19**, after 2 h incubation at 37 °C, the long AF were detected in the water by AFM, confirming the presence and release of AF from the coacervate capsule to the bulk of water. However, for the sample treated with SGF, the AFM image did not show any

evidence of the existence of AF but short fragments, indicating the complete digestion of AF. As HA has adhesive properties⁶⁴, the hypothesis that we suggest for the protective effect of microparticles is that they can adhere to the mucosa, and then HA and AF may synergistically contribute to the ulcer gastric protective properties of the coacervate microparticles, highlighting them as systems with efficient biological activity for gastric ulcer protection and possible therapeutic use.

59 Savarino, E. et al. *Drugs for improving esophageal mucosa defense: where are we now and where are we going?* *Ann Gastroenterol* 30, 585-591, doi:10.20524/aog.2017.0187 (2017).

60 Dovedytis, M., Liu, Z. J. & Bartlett, S. *Hyaluronic acid and its biomedical applications: A review.* *Engineered Regeneration* 1, 102-113, doi:https://doi.org/10.1016/j.engreg.2020.10.001 (2020).

61 Fan, Y., Choi, T.-H., Chung, J.-H., Jeon, Y.-K. & Kim, S. *Hyaluronic acid-cross-linked filler stimulates collagen type I and elastic fiber synthesis in skin through the TGF- β /Smad signaling pathway in a nude mouse model.* *Journal of Plastic, Reconstructive & Aesthetic Surgery* 72, 1355-1362, doi:https://doi.org/10.1016/j.bjps.2019.03.032 (2019).

62 Kathem, R. A. et al. *Antinociceptive and Anti-Inflammatory Activity of Bovine Milk-Derived β -Lactoglobulin: Study of Combined Effect with Tramadol or Indomethacin.* *Global Journal of Pharmacology* 8, 444-456, doi:10.5829/idosi.gjp.2014.8.3.1119 (2014).

63 Rosaneli, C. F., Bighetti, A. E., Antônio, M. A., Carvalho, J. E. & Sgarbieri, V. C. *Protective Effect of Bovine Milk Whey Protein Concentrate on the Ulcerative Lesions Caused by Subcutaneous Administration of Indomethacin.* *Journal of Medicinal Food* 7, 309-314, doi:10.1089/jmf.2004.7.309 (2004).

64 Marinho, A., Nunes, C. & Reis, S. *Hyaluronic Acid: A Key Ingredient in the Therapy of Inflammation.* *Biomolecules* 11, doi:10.3390/biom11101518 (2021).

REVIEWERS' COMMENTS

Reviewer #1 (Remarks to the Author):

The authors have addressed the review comments well. There is one question where further details are required.

The original question was:

At line 313, the films formed are described as being smeared i.e. one phase on top of another. Using what instrument and what method? Can the method be described adequately so that another scientist might repeat the method?

The answer was:

The films were prepared simply by placing one phase on top of the other without using any instrument or particular methods.

Was a pipette used to dispense the liquid phase? Was the solution poured from a beaker? How was the layering performed?

Reviewer #2 (Remarks to the Author):

This is a multifaceted manuscript that will likely capture the interests of material scientists as well as the biopharmaceutical community. Therefore I support publication in Nature Communications.

I am impressed with both the extensive nature of the reviews as well as the thorough responses from the authors. The manuscript has certainly been improved significantly. The community came together to improve the presentation of a lovely interdisciplinary contribution.

Reviewer #3 (Remarks to the Author):

The authors have done an admirable job responding to the significant feedback from reviewers. The authors have updated the manuscript with additional data that support many of their findings and have modified the text to include more details of the system. The additional data does support the importance of electrostatic interactions in forming the condensed phase (e.g. pH study data and ionic strength dependence), however this additional evidence does not support their description of the process as complex conservation, or the liquid-liquid phase separation of oppositely charged species. There is no evidence that the second phase formed is a liquid - in fact the authors show that the AF and polysaccharide form a gel. I am not suggesting that the authors do more experiments, but rather are careful in the language used to describe of the physical process they observe. This may still be a phase

transition (or simply network formation) that results in the formation of a solid or a gel. With these textual changes to modify the language regarding 'coacervation' (complex or not) and 'liquid-liquid phase separation', I am supportive of the publication of this work.

Manuscript NCOMMS-22-43643A

Reviewers' comments:

Reviewer #1 (Remarks to the Author):

The authors have addressed the review comments well.

We thank the reviewer for the positive comments on our manuscript; we have followed his/her last suggestion to revise the manuscript.

There is one question where further details are required.

The original question was:

At line 313, the films formed are described as being smeared i.e. one phase on top of another.

Using what instrument and what method? Can the method be described adequately so that another scientist might repeat the method?

The answer was:

The films were prepared simply by placing one phase on top of the other without using any instrument or particular methods.

Was a pipette used to dispense the liquid phase? Was the solution poured from a beaker? How was the layering performed?

The films were formed by gently smearing 1 ml of AF phase with a pipette on top of 1 ml of HA phase on a Petri dish. After 2 h, the formed film was thoroughly washed with Milli-Q water and dried at 40 °C.

Reviewer #2 (Remarks to the Author):

This is a multifaceted manuscript that will likely capture the interests of material scientists as well as the biopharmaceutical community. Therefore I support publication in Nature Communications.

I am impressed with both the extensive nature of the reviews as well as the thorough responses from the authors. The manuscript has certainly been improved significantly. The community came together to improve the presentation of a lovely interdisciplinary contribution.

We thank the reviewer for her/his positive words, and reciprocally, we enjoyed the constructive input of the reviewer in enhancing the quality of this work.

Reviewer #3 (Remarks to the Author):

The authors have done an admirable job responding to the significant feedback from reviewers. The authors have updated the manuscript with additional data that support many of their findings and have modified the text to include more details of the system. The additional data does support the importance of electrostatic interactions in forming the condensed phase (e.g. pH study data and ionic strength dependence), however this additional evidence does not support their description of the process as complex coacervation, or the liquid-liquid phase separation of oppositely charged species. There is no evidence that the second phase formed is a liquid - in fact the authors show that the AF and polysaccharide form a gel. I am not suggesting that the authors do more experiments, but rather are careful in the language used to describe of the physical process they observe. This may still be a phase transition (or simply network formation) that results in the formation of a solid or a gel. With these textual changes to modify the language regarding 'coacervation' (complex or not) and 'liquid-liquid phase separation', I am supportive of the publication of this work.

We are grateful that the reviewer appreciates our extensive work in revising the paper based on her/his comments.

In the previous round of review, the reviewer doubted that the materials are formed via complex coacervation or any type of liquid-liquid phase separation since *it is not clear that in fact a second liquid phase, at equilibrium with a dilute phase, is forming or rather if the two oppositely charged biopolymers are simply aggregating or precipitating*. In the new review report, although the reviewer appreciated that we clearly revealed the importance of electrostatic interactions in forming the condensed phase by further experiments on pH and ionic strength dependence (as the major parameters on complex coacervation), she/he did neglect the experiments that we have performed to clearly depict the formation of the second liquid phase, at equilibrium with a dilute phase. Interestingly she/he again mentioned *There is no evidence*

that the second phase formed is a liquid - in fact the authors show that the AF and polysaccharide form a gel.

During AF and HA coacervation in this work, the second liquid phase, a dilute phase, is formed.

Supplementary Fig. 2 exhibits the bulk coacervation of AF and HA, where the coacervate and dilute phase (equilibrium solution) are clearly distinguishable. The equilibrium solution was transparent and had a very low viscosity, different from the two precursors.

Supplementary Fig. 2. Bulk coacervation of AF (2 wt.%) and HA (1 wt.%) at pH 3. The two resulting phases after coacervation, i.e., coacervate and diluted phases, are visible.

Coacervation is an associative liquid-liquid phase separation (LLPS) phenomenon that occurs through electrostatic interaction between oppositely charged macromolecular building blocks, such as proteins, polysaccharides and polymers in solution, resulting in a colloidal rich phase, the coacervate, and a remaining biomacromolecule-depleted phase in equilibrium with the denser coacervate^{22,23}.

Indeed, the coacervation between proteins and polysaccharides is a very well-known phenomenon in the field since the pioneering work of Bungenberg de Jong et al. on gelatin-gum Arabic complex coacervation in the 1920-40s (*Protein/polysaccharide complexes and coacervates in food systems, Advances in Colloid and Interface Science 167 (2011) 63–70*) and reported by many different works (*Complex coacervation of proteins and anionic polysaccharides, Current Opinion in Colloid*

& *Interface Science* 9 (2004) 340 – 349; *Composition and Rheological Properties of β -Lactoglobulin/Pectin Coacervates: Effects of Salt Concentration and Initial Protein/Polysaccharide Ratio*, *Biomacromolecules* 2007, 8, 3, 992–997, *Protein-Selective Coacervation with Hyaluronic Acid*, *Biomacromolecules* 2014, 15, 3, 726–734).

The reviewer is correct in the sense that the final separated phase may acquire solid-like properties vs. liquid-like and indeed, it is well known that the coacervate/condensate/separated phase may mature into a solid-like material: see Hyman et al. *Science* 2020.

In order to clarify this point further and avoid any residual, possible ambiguity, we have added the following text:

LLPS and coacervates can eventually undergo aging and lead to a liquid-to-solid transition (LST) and precipitation²⁶. Although coacervation can take place in the absence of precipitation, precipitation induced by a pH change is always preceded by coacervation²⁷.

The highlighted text is added to the manuscript accordingly for more clarification. We could, of course, replace the term “coacervates” with the more general term “condensates” which does not require the presence of electrostatic interactions; However, precisely because we have demonstrated the key role of electrostatic (in response to the same reviewer previous comments) in the associative process described, we prefer to keep the terminology of coacervation which is more precise and accurate, along with liquid-liquid phase separation (as the main initiating event), in agreement with most of the literature we are aware of.

²⁶ Jawerth, L. et al. *Protein condensates as aging Maxwell fluids*. *Science* 370, 1317-1323, doi:doi:10.1126/science.aaw4951 (2020).

²⁷ Comert, F., Malanowski, A. J., Azarikia, F. & Dubin, P. L. *Coacervation and precipitation in polysaccharide–protein systems*. *Soft Matter* 12, 4154-4161, doi:10.1039/C6SM00044D (2016).

Reviewer #3 (Remarks to the Author):

The authors have continued to improve their manuscript and have added useful references and context to indicate that coacervation or precipitation can occur upon forming polyelectrolyte complexes (and additionally that liquid coacervates can age to form gels or solids as well). While I disagree with their characterization of the second, dense phase that is formed, I do not feel that this reviewer's desire to accurately describe this phase is sufficient to prevent publication of the manuscript.

I am fine with the manuscript to be published as it is now but would encourage the authors to:

(1) Consider the context of the study they cite regarding the claim that coacervation always precedes precipitation. In this cited study (ref 27) the authors are performing a pH titration on a single mixed sample rather than mixing samples at discrete pH values (as is done in this manuscript). It seems to reason then that if the authors of the original study (ref 27) had simply mixed at the pH that resulted in precipitation they could have observed direct precipitation rather than coacervation followed by precipitation. So, I am not sure this provides the same strong evidence that the authors claim to be analogous to their experimental situation.

(2) Consider if the floating, irregularly shaped, opaque dense phase (labeled 'coacervate' in Supplementary Figure 2) suspended in a dilute phase is likely the equilibrium structure for two immiscible liquids.